

# Choosing between post-processing precipitation forecasts or chaining several uncertainty quantification tools in hydrological forecasting systems

Emixi Sthefany Valdez[1], François Anctil[1], and Maria-Helena Ramos[2]

[1]Dept. of Civil and Water Engineering, Université Laval, 1065 Avenue de la Médecine, Québec, Canada
[2]Université Paris-Saclay. INRAE, UR HYCAR, 1 Rue Pierre-Gilles de Gennes, 92160 Antony, France

**Correspondence:** Emixi Valdez (emixi-sthefany.valdez-medina.1@ulaval.ca)

**Abstract.** This study aims to decipher the interactions of a precipitation post-processor and several other tools for uncertainty quantification implemented in a hydrometeorological forecasting chain. We make use of four hydrometeorological forecasting systems that differ by how uncertainties are estimated and propagated. They consider the following sources of uncertainty: A) forcing, B) forcing and initial conditions, C) forcing and model structure, and D) forcing, initial conditions, and model structure. For each system's configuration, we investigate the reliability and accuracy of post-processed precipitation forecasts in order to evaluate their ability to improve streamflow forecasts for up to seven days of forecast horizon. The evaluation is carried out across 30 catchments in the Province of Quebec (Canada) and over the 2011-2016 period. Results are compared using a multicriteria approach, and the analysis is performed as a function of lead time and catchment size. The results indicate that the precipitation post-processor resulted in large improvements in the quality of forecasts with regard to the raw precipitation forecasts. This was especially the case when evaluating relative bias and reliability. However, its effectiveness in terms of improving the quality of hydrological forecasts varied according to the configuration of the forecasting system, the forecast lead time, and the catchment size. The combination of the precipitation post-processor and the quantification of uncertainty from initial conditions showed the best results. When all sources of uncertainty were quantified, the contribution of the precipitation post-processor to provide better streamflow forecasts was not remarkable and, in some cases, it even deteriorated the overall performance of the hydrometeorological forecasting system. Our study provides an in-depth investigation on how improvements brought by a precipitation post-processor to the quality of the inputs to a hydrological forecasting model can be cancelled along the forecasting chain, depending on how the hydrometeorological forecasting system is configured and on how the other sources of hydrological forecasting uncertainty (initial conditions and model structure) are considered and accounted for. This has implications for the choices users might make when designing new or enhancing existing hydrometeorological ensemble forecasting systems.

## 1 Introduction

Reliable and accurate hydrological forecasts are critical to several applications such as preparedness against flood-related casualties and damages, water resources management, and hydropower operations (Alfieri et al., 2014; Bogner et al., 2018; Boucher





et al., 2012; Cassagnole et al., 2021). Accordingly, different methods have been developed and implemented to represent the

errors propagated throughout the hydrometeorological forecasting chain and improve operational forecasting systems (Zappa et al., 2010; Pagano et al., 2014; Emerton et al., 2016). The inherent uncertainty of hydrological forecasts stems from three main sources: 1) the hydrological model structure and parameters, 2) the initial hydrological conditions, and 3) the meteorological forcing (Schaake et al., 2007; Thiboult et al., 2016). Two main philosophies are generally adopted in the literature to quantify those uncertainties: statistical methods and ensemble-based methods (e.g., Boelee et al., 2019). The latter is increasingly being

used operationally or pre-operationally (Olsson and Lindström, 2008; Coustau et al., 2015; Addor et al., 2011; Demargne et al., 2014) for short (up to 2-3 days), medium (up to 2 weeks), and extended (sub-seasonal and seasonal) forecast ranges (Pappenberger et al., 2019). It relies on issuing several members of an ensemble (possible future evolution of the forecast variable) or combining many scenarios of model structure and/or parameters, catchment descriptive states, and forcing data. Probabilistic guidance can be generated from the ensemble and provide useful information about forecast uncertainty to users (Zappa et al.,

2019). Additionally, the contribution of each component of uncertainty quantification in a forecasting system can be assessed, which is not possible with the statistical philosophy since it evaluates total uncertainty (Demirel et al., 2013; Kavetski et al., 2006).

To represent the hydrological model structure and parameters uncertainty, the multimodel framework has become a viable solution (Velázquez et al., 2011; Seiller et al., 2012; Thiboult and Anctil, 2015; Thiboult et al., 2016). Model structure uncer-

tainty has proven to be dominant compared to uncertainty in parameter estimation (Poulin et al., 2011; Gourley and Vieux, 2006; Clark et al., 2008) or to solely increasing the number of members of an ensemble (Sharma et al., 2019). Regarding the quantification of the initial conditions uncertainty in hydrological forecasting, many data assimilation (DA) techniques have been proposed (Liu et al., 2012). The most common DA methods in hydrology are the particle filter (e.g., DeChant and Moradkhani, 2012; Thirel et al., 2013), the ensemble Kalman Filter (e.g., Rakovec et al., 2012) and its variants (e.g., Noh et al., 2014).

The use of DA techniques usually enhance performance, comparatively to an initial model simulation without DA, especially in the short range and early medium range. (Bourgin et al., 2014; Thiboult et al., 2016; DeChant and Moradkhani, 2011).

Meteorological forcing uncertainty is primarily tackled by ensembles of numerical weather prediction (NWP) model outputs (Cloke and Pappenberger, 2009), generated by running the same model several times from slightly different initial conditions and/or using stochastic patterns to represent model variations. Notwithstanding substantial improvements made in NWP, it is

still a challenge to represent phenomena at sub-basin scales correctly, particularly convective processes (Pappenberger et al., 2005). Meteorological models also face difficulties predicting the magnitude, time, and location of large precipitation events, which are dominant flood-generating mechanisms in small and large river basins. Systematic biases affecting the accuracy and reliability of numerical weather model outputs cascade into the hydrological forecasting chain and may be amplified on the streamflow forecasts. Consequently, a meteorological post-processor (sometimes named pre-processor in hydrology as it refers

to corrections to hydrological model inputs) is nowadays an integral part of many hydrological forecasting systems (Schaake et al., 2007; Gneiting, 2014; Yu and Kim, 2014; Anghileri et al., 2019), especially in the longer forecast ranges, for which the meteorological model resolution affecting the simulation of precipitation processes is generally coarser than the one used for short and medium ranges (Crochemore et al., 2016; Lucatero et al., 2018; Monhart et al., 2019). The degree of complexity of





these precipitation post-processing techniques varies from simple systematic bias correction to more sophisticated techniques
involving conditional distributions (see Li et al. (2017) and Vannitsem et al. (2020) for reviews).

While some studies based on medium-range forecasts suggested important improvements in the quality of streamflow forecasts after post-processing precipitation forecasts (Aminyavari and Saghafian, 2019; Yu and Kim, 2014; Cane et al., 2013), others indicate that improvements in precipitation and temperature forecasts do not always translate into better streamflow forecasts, at least not proportionally (Verkade et al., 2013; Zalachori et al., 2012; Roulin and Vannitsem, 2015). It is suggested
that precipitation post-processing application should be carried out only when the inherent hydrological bias is eliminated or at least reduced. Otherwise, its value may be underestimated (Kang et al., 2010; Sharma et al., 2018). In other words, if the hydrometeorological forecasting system does not have components to estimate the other sources of uncertainties in the forecasting chain (e.g., model structure and initial conditions uncertainty), a meteorological post-processor alone does not guarantee improvements in the hydrological forecasts. Therefore, hydrological post-processors (targetting bias correction of
hydrological model outputs) are often advocated to deal with the bias and the underdispersion of ensemble forecasts caused by the partial representation of forecast uncertainty (Boucher et al., 2015; Brown and Seo, 2013). Hydrological post-processing alone has shown to be a good alternative for improving forecasting performance (Zalachori et al., 2012; Sharma et al., 2018), but it cannot compensate for weather forecasts that are highly biased (Abaza et al., 2017; Roulin and Vannitsem, 2015). For instance, precipitation biases towards underestimation could lead to missing events since the hydrological model will fail to
exceed flood thresholds. In this regard, a common question in the operational hydrometeorological forecasting community is whether post-processing efforts should be applied to weather forecasts, hydrological forecasts, or both.

Recently, Thiboult et al. (2016, 2017) discuss the need for post-processing model outputs when the main sources of uncertainty in a hydrological forecasting chain are correctly quantified. Post-processing model outputs adds a cost to the system, may not substantially improve the outputs, and may even add additional sources of uncertainty to the whole forecasting process.
For example, after the application of some statistical techniques, the spatio-temporal correlation (Clark et al., 2004; Schefzik et al., 2013) and the coherence (when forecasts are at least as skillful as climatology, (Gneiting, 2014)) of the forecasts can be destroyed. Nevertheless, quantifying "all" uncertainties may limit the operational applicability of forecasting systems, especially when there are limitations in data and computational time management (Boelee et al., 2019; Wetterhall et al., 2013). Considering several sources of uncertainty in ensemble hydrometeorological forecasting inevitably implies an increase in the
number of simulations. A larger number of members in an ensemble forecasting system could allow a closer representation of the full marginal distribution of possible future occurrences and yield better forecasts (Houtekamer et al., 2019). However, there is also additional uncertainty associated with the assumptions made in creating a larger ensemble. As Beven (2016) pointed out: "There is a lot of uncertainty about uncertainty estimation." If the methodologies implemented to simulate the sources of forecast error and quantify different uncertainty sources are inappropriate, even a large ensemble size could be under or
overdispersive and thus not represent the total predictive uncertainty accurately (Boelee et al., 2019; Buizza et al., 2005).

Trade-offs are inevitable when defining the configuration of an ensemble hydrometeorological forecasting system to be implemented in an operational context. The traditional sources of uncertainty (i.e., initial conditions, hydrological model structure and parameters, and forcings) are rarely considered fully and simultaneously. There are still gaps in understanding the





way they interact with the dominant physical processes and flow-generating mechanisms that operate on a given river basin

(Pagano et al., 2014). Meanwhile, operational weather forecasts have constantly been improving and will continue to evolve in the future. For example, improvements have been made on three of the key characteristics of the ensemble weather forecasts of the European Centre for Medium-Range Weather Forecasts (ECMWF): vertical and horizontal resolution, forecast length, and ensemble size (Buizza, 2019; Buizza and Leutbecher, 2015; Palmer, 2019). In May 2021, an upgrade of ECMWF's Integrated Forecasting System (IFS) has introduced single precision for high-resolution and ensemble forecasts, which is expected to

increase forecast skill across different time ranges. Therefore, it is relevant for hydrologists to better understand in which circumstances they can directly use NWPs outputs without compromising hydrological forecasting performance. It is necessary to evaluate how each component of a forecasting system interacts with the other and to understand how they contribute to forecast performance. This may give clues of where to focus investments: should we favor a sophisticated system accounting for many sources of uncertainty or a simpler one endowed with post-processing for bias correction? Notably, several studies

highlight in unison the need for further research regarding the incorporation of precipitation post-processing techniques and the evaluation of their interaction with the other components of the hydrometeorological modeling chain for diverse hydroclimatic conditions (Wu et al., 2020).

This study aims to identify in which circumstances the implementation of a post-processor of precipitation forecasts would significantly improve hydrological forecasts. For this, we investigate the interactions among several state-of-the-art tools for

uncertainty quantification implemented in a hydrometeorological forecasting chain. More specifically, the following questions are addressed:

– Does precipitation post-processing improve streamflow forecasts when dealing with a forecasting system that fully or partially quantifies other sources of uncertainty?

– How does the performance of different uncertainty quantification tools compare?

– How does each uncertainty quantification tool contribute to improving streamflow forecast performance across different lead times and catchment sizes?

We created four hydrometeorological forecasting systems that differ by how uncertainties are estimated and propagated. They consider the following sources of uncertainty: A) forcing, B) forcing and initial conditions, C) forcing and model structure, and D) forcing, initial conditions, and model structure. We considered the ECMWF ensemble precipitation forecast over

the period 2011-2016 and up to 7 days of forecast horizon, seven hydrological lumped conceptual models, and the Ensemble Kalman filter as tools for uncertainty quantification. These three tools represent the forcing, model structure, and initial conditions uncertainties, respectively. We investigated their performance across 30 catchments in the Province of Quebec (Canada) for each system. Precipitation forecasts are post-processed by applying the Censored, Shifted Gamma distribution proposed by Scheuerer and Hamill (2015).

This paper is structured as follows: Sect. 2 presents the methods, data sets, and case study. Section 3 presents the results, followed by a discussion in Sect. 4, and finally the conclusions in Sect. 5.



## 2 Methods and case study

### 2.1 Tools for uncertainty quantification

The ensemble forecasting approach allows to distinguish the part of uncertainty that comes from different sources. A specific
source of uncertainty can be tracked through the modeling chain and along lead times (Boelee et al., 2019). Following Thiboult
et al. (2016), we created four ensemble prediction systems that differ on how hydrometeorological uncertainties are quantified
(Table 1). Ensembles were built from the HOOPLA (HydrOlOgical Prediction LAboratory) modular framework (Thiboult
et al., 2018), an automatic software that allows to carry out model calibration and obtain hydrological simulations and forecasts
at different time steps. They consider uncertainty coming from: System A) forcing, System B) forcing and initial conditions,
System C) forcing and model structure, and System D) forcing, initial conditions, and model structure. Each source contributes
a number of members, from 7 to 50, to the forecasting system when it is turned on, as shown in Table1. In systems for which
modeling uncertainty is not considered (i.e., A and B), only one model is used and it is the one presenting median performance
during calibration. As shown in Table 1, all systems quantify the forcing (in our case, precipitation) uncertainty. Raw and
post-processed ensemble precipitation forecasts drive the hydrological model(s).
In the following sections, we describe each tool that quantifies a different source of uncertainty as applied in this study. We
include each technique's conceptual aspects and the reasons behind their selection.

**Table 1.** Forecasting systems A, B, C and D of the study and total number of members of the resulting ensemble streamflow forecast. On
(Off) indicates when uncertainty is (is not) quantified with the help of ensemble members.

| | Systems | | | |
|---|---|---|---|---|
| **Source [number of members when 'On']** | **A** | **B** | **C** | **D** |
| Forcing (precipitation) [50 members] | On | On | On | On |
| Initial conditions [50 members] | Off | On | Off | On |
| Model structure [7 hydrological models/members] | Off | Off | On | On |
| **Nb of members** | **50** | **2,500** | **350** | **17,500** |

### 2.1.1 Precipitation forecast uncertainty: ensemble forecast

Forcing uncertainty is characterized by precipitation ensemble forecasts issued by the European Center for Medium-Range
Weather Forecasts (ECMWF), downloaded through the TIGGE database for the 2011-2016 period. The set consists of 50
exchangeable members generated from the perturbation of the most accurate estimate of the current atmospheric condition,
termed the control forecast. During the period considered, those perturbations were simulated using the combination of the
singular vector and data assimilation techniques (Buizza and Palmer, 1995; Buizza et al., 2008). Meteorological model uncer-
tainties were taken into account using stochastic schemes (Buizza et al., 1999; Shutts, 2005).





The ECMWF ensemble is issued twice a day (0:00 and 12:00 UTC) with a forecast horizon extending up to 15 days at a
six-hour time step. Forecasts are archived using a reduced Gaussian grid that is transformed to the latitude-longitude system
during TIGGE extraction by bilinear interpolation (Gaborit et al., 2013). The database was originally stored into two spatial
resolutions: $0.25°$ (in either direction) for lead times up to 10 days and $0.50°$ for lead times between 10 and 15 days. However,
the declining resolution on day 10 is no longer experienced since March 2016. In this study, the resolution was reduced to $0.1°$,
which has proven to be suitable to the selected catchments (Thiboult et al., 2016).

Forecasts issued at 12:00 UTC, for a maximum forecast horizon of 7 days, were considered here. Forecasts and observations
were temporally aggregated to a daily time step and spatially averaged to the catchment scale to match the common HOOPLA
framework of the hydrological models. In order to isolate the effect of precipitation in this study, observed air temperatures
were used instead of the forecast ones. This allows us to focus on changes in streamflow forecast performance attributed only
to precipitation post-processing, which is typically the most challenging variable to simulate and the one that mostly impacts
hydrological forecasting (Hagedorn et al., 2008).

### 2.1.2 Precipitation post-processor: Censored, shifted gamma distribution (CSGD)

The ECMWF ensemble precipitation forecasts were post-processed over the 2011-2016 period following a simplified variant
of the CSGD method proposed by Scheuerer and Hamill (2015). The CSGD is based on a complex heteroscedastic, nonlinear
regression model conceived to address the peculiarities of precipitation (e.g., its intermittent and highly skewed nature and
its typically large forecast errors). This method yields full predictive probability distributions for precipitation accumulations
based on ensemble model output statistics (EMOS) and censored, shifted gamma distributions. We selected the CSGD method
because it has broadly outperformed other established post-processing methods, especially in processing intense rainfall events
(Scheuerer and Hamill, 2015; Zhang et al., 2017). Moreover, its relative impact on hydrological forecasts has already been
assessed at different scales and in various hydroclimatic conditions (Bellier et al., 2017a; Scheuerer et al., 2017).

In this study, the original version of the CSGD method was adapted because the ensemble statistics had to be determined
on the average catchment rainfall rather than using the information from the neighboring grid points. Figure 1 identifies the
different stages necessary to apply the method. Briefly, the application of the CSGD was accomplished as follow:

1. Errors in the ensemble forecasts climatology were corrected via the Quantile Mapping (QM) procedure advocated by
   Scheuerer and Hamill (2015). The QM method adjusts the cumulative distribution function (CDF) of the forecasts onto
the observations. In this version, the quantiles were estimated from the empirical CDFs. See Scheuerer and Hamill (2015)
   for details about the procedure and extrapolations beyond the extremes of the empirical CDFs.

2. The corrected forecasts were condensed into statistics, used as predictors to drive a heteroscedastic regression model.
   The predictors were the ensemble mean $(\bar{f})$, the ensemble probability of precipitation $(POP_f)$, and the ensemble mean
   difference $(MD_f)$. The latter is a measure of the forecast spread. In regression methods, the skill of post-processed
forecasts is sensitive to the choice of predictors and their number (Brown et al., 2013; Van Schaeybroeck and Vannitsem,
   2011). Nevertheless, Scheuerer and Hamill (2015) demonstrated that adding nonlinear and heteroscedastic components



to a regression model plays a more decisive role in forecasting performance than the selected predictors, especially in low predictability situations (e.g., longer lead times). This statement was corroborated by Zhang et al. (2017), who evaluated a simple version of the CSGD method (one predictor) and a more complex one (three predictors) against the mixed-type meta-Gaussian distribution.

3. The CSGD model with mean ($\mu$), standard deviation ($\sigma$), and shift ($\delta$; it controls $POP_f$=0) parameters was fitted to the climatological distribution of observations to establish the parameters for the unconditional CSGD ($\mu_{cl}$, $\sigma_{cl}$, $\delta_{cl}$). The ensemble statistics from step 1 and the unconditional CSGD parameters were linked to the CSGD model via:

$$\mu = \frac{\mu_{cl}}{a_1} log1p \left[ expm1(a_1)(a_2 + a_3 POP_f + a_4 \bar{f}) \right] \tag{1}$$

$$\sigma = \sigma_{cl} \left( b_1 \sqrt{\frac{\mu}{\mu_{cl}}} + b_2 MD_f \right) \tag{2}$$

$$\delta = \delta_{cl} \tag{3}$$

where $log1p(x) = log(1 + x)$, and $expm1(x) = exp(x) - 1$. The $\delta$ parameter accounts for the probability of zero precipitation. Both the unconditional CSGDs and the regression parameters are fitted by minimizing a closed form of the continuous ranked probability score (CRPS). We refer to Scheuerer and Hamill (2015) for more details about the equations, model structure and fitting.

Considering that precipitation (and particularly intense events) does not have a temporal autocorrelation (memory) as strong as streamflow (Li et al., 2017), we adopted the standard leave-one-year-out cross-validation approach to estimate the CSGD climatological and regression model parameters. They were fitted for each month and lead time, using a training window of approximately three months (all forecast and observations from 90 days around the 15th day of the month under consideration), resulting in a training sample size of 91 X 5 pairs of observations and forecasts.

The CSGD method yields a predictive distribution for each catchment, lead time, and month. This distribution allows one to make a sample and construct an ensemble of any desired size $M$. As comparing ensembles of different sizes may induce a bias (Buizza and Palmer, 1998; Ferro et al., 2008), we drew ensembles of the same size as the raw forecasts ($M = 50$). Similar to Scheuerer and Hamill (2015), we sampled the full distribution by choosing the quantiles with level $\alpha_m = (m - 0.5/M)$ for $m = 1, ..., M$, which correspond to the optimal sample of the predictive distribution that minimizes the CRPS (Bröcker, 2012).

### 2.1.3 Reordering method: Ensemble copula coupling

EMOS procedures, such as the CSGD method, destroy the spatio-temporal and intervariable correlation of the forecasts. Many studies stressed the importance of correctly reconstructing the dependence structures of weather variables for hydrological ensemble forecasting (Bellier et al., 2017a; Scheuerer et al., 2017; Verkade et al., 2013). In this study, we use the ensemble copula coupling (ECC) (Schefzik et al., 2013) to address this issue. The ECC is a nonparametric and straightforward reordering





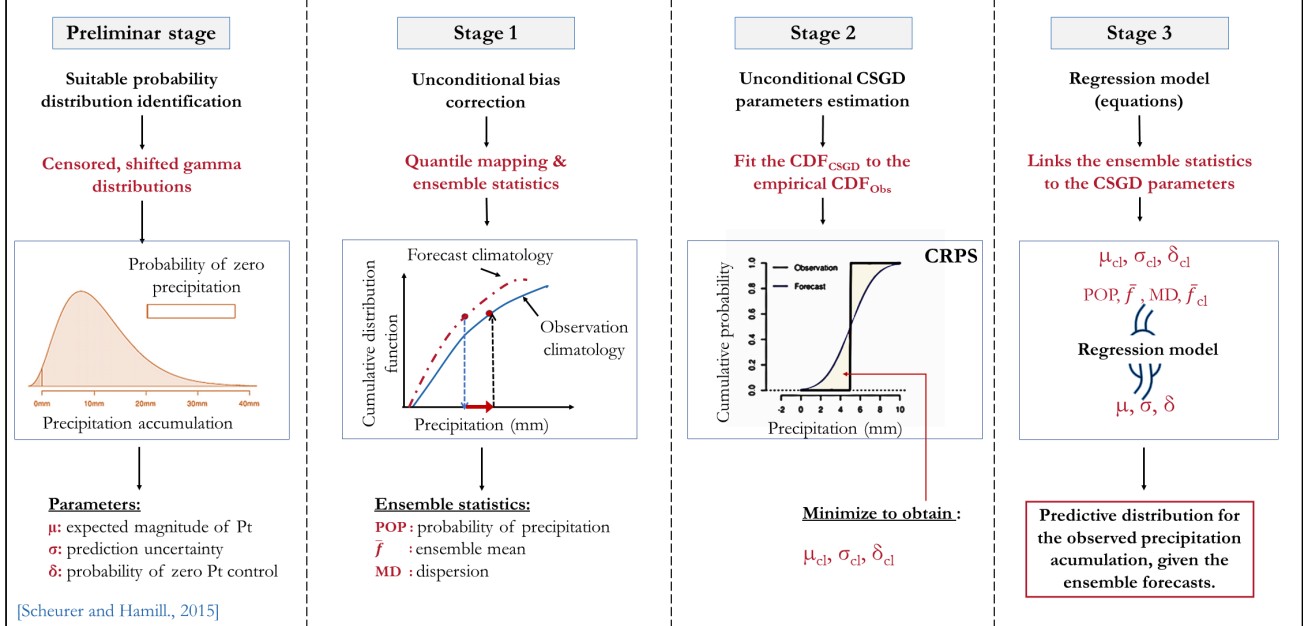

**Figure 1.** Stages of the CSGD precipitation post-processor.

technique. As such, it relaxes the constraint of parametric assumptions (difficult for variables like precipitation) and high-dimensional cases (Gneiting, 2014). This technique uses the raw ensemble forecasts trajectories to identify the dependence template. Therefore, the post-processed ensemble should have the same number of members as the raw ensemble. The ECC reasoning lies in the fact that the physical model can adequately represent the covariability between the different dimensions

215 (i.e., space, time, variables). The predictive distribution sample with equidistant quantiles with levels $\alpha_m = (m - 0.5/M)$ is reordered so that the rank order structure is the same as the raw ensemble values. We refer to Schefzik et al. (2013) for more details about the mathematical framework underlying the method.

### 2.1.4 Initial conditions. uncertainty: Ensemble Kalman Filter

The Ensemble Kalman Filter (Evensen (2003), EnKF) is used to provide model state at each time step. The EnKF is a sophis-

220 ticated sequential and probabilistic data assimilation technique that relies on a Bayesian approach. It estimates the probability density function of model states conditioned by the distribution of observations. In this study, we use the same hyperparameters (EnKF settings) than Thiboult et al. (2016). They were identified after rigorous testing carried out by Thiboult and Anctil (2015) using model performance on reliability and bias as criteria of selection over the same hydrologic region as the present study.



The EnKF performance is highly sensitive to its hyperparameters (Thiboult and Anctil, 2015), which represent the uncertainty around model inputs and outputs. For precipitation, we used 50 % standard deviation of the mean value with a gamma law. For streamflow and temperature, we used 10 % and 2 °C standard deviation with normal distribution, respectively.

At every time step, the EnKF is tuned to optimize reliability and accuracy, per catchment and hydrological model, following two principal stages:

1. **Forecasting**: $N$ forcing scenarios are propagated using the hyperparameters through the model to generate $N$ members of state variables from the prior estimate of the state ($X_t^-$, also called background or predicted state). From this ensemble of state variables, the model's error covariance matrix ($P_t$, the difference between the true state and the individual hydrological model realizations) is computed and used to calculate the Kalman gain ($K_t$). Then, the Kalman gain is calculated from $P_t$ and $R_t$ (the covariance of observation noise) according to a weighting coefficient used to update the states of the model. The Kalman gain is mathematically represented as:

$$K_t = P_t H^T (H P_t H^T + R_t)^{-1},$$ (4)

where the $t$ indices refer to the time and $H$ is the observation function that relates the state vectors and the observations.

2. **Update (analysis)**: once an observation becomes available ($z_t$), the state variables ($X_t^+$) are updated as a combination of the prior knowledge of the states ($X_t^-$), the Kalman gain ($K_t$), and the innovation (i.e., the difference between the observed and the prior simulated streamflow).

$$X_t^+ = X_t^- + K_t(z_t - H X_t^-),$$ (5)

A full description of the EnKF scheme applied in this study is provided in Thiboult and Anctil (2015).

The work of Thiboult and Anctil (2015) demonstrated that EnKF performance is not as sensitive to the number of members as it is to the hyperparameters (at least for the catchments and models used in this study). Ensembles of 25 and 200 members presented similar performances. Therefore, we opted for 50 members as a trade-off between computational cost and stochastic errors when sampling the marginal distributions of the state variables.

### 2.1.5 Hydrological uncertainty: hydrological models, snow module, and evapotranspiration

To consider model structure and parametrization uncertainties, we use seven of the 20 lumped conceptual hydrological models available in the HOOPLA framework. Keeping in mind parsimony and diversity as criteria (different contexts, objectives, and structures), Seiller et al. (2012) have selected these 20 models, expanding from an initial list established by Perrin (2000).

To maximize the benefits from the multimodel approach, with the constraint of low computational time and data management, we opted for seven models with particular attention paid to how they represent flow production, draining, and routing





processes. The structures of the selected models vary from 6 to 9 free parameters and from 2 to 5 water storage elements. All models include a soil moisture accounting storage and at least one routing process. Diversity can be a useful feature for

forecasting events beyond the range of responses observed during model calibration (Beven, 2012; Beven and Alcock, 2012) and for catchments that present strong heterogeneities (Kollet et al., 2017). Table 2 summarizes the main characteristics of the lumped models and identifies the original models from which they were derived.

**Table 2.** Main characteristics of the seven lumped models. Modified from Seiller et al. (2012).

| Model | No. of Parameters | No. of reservoirs | Derived from |
|:---:|:---:|:---:|---|
| **M1** | 9 | 2 | CEQUEAU (Girard et al., 1972) |
| **M2** | 9 | 3 | HBV (Bergström and Forsman, 1973) |
| **M3** | 7 | 3 | IHACRES (Jakeman et al., 1990) |
| **M4** | 6 | 4 | MORDOR (Garçon, 1999) |
| **M5** | 8 | 4 | PDM (Moore and Clarke, 1981) |
| **M6** | 9 | 5 | SACRAMENTO (Burnash et al., 1973) |
| **M7** | 8 | 4 | XINANJIANG (Zhao et al., 1980) |

All hydrological models are individually coupled with the CemaNeige snow accounting routine (Valéry et al., 2014). This two-parameter module estimates the amount of water from melting snow based on a degree-day approach. It relies on a

spatial discretization of the basin into five altitudinal layers of equal area. Fed with total precipitation, air temperature, and elevation data, CemaNeige separates the solid precipitation fraction from the liquid fraction and stores it in a conceptual reservoir (snowpack). The model simulates two internal state variables of the snowpack for each zone: the thermal inertia of the snowpack (Ctg [-], higher values indicate later snowmelt) and a degree-day melting factor (Kf [$mm°C^{-1}$], higher values indicate a faster rate of snowmelt). The latter determines the elapsed melt blade that will be added to the hydrological model.

These parameters are optimized for each model.

All hydrological models were forced with the same input data: daily precipitation and ETP based on catchment's air temperature and the extraterrestrial radiation (Oudin et al., 2005).

To calibrate the hydrological models, we computed the modified King-Gupta Efficiency (KGEm) as objective function (Gupta et al., 2009; Kling et al., 2012) and used the SCE as the automatic optimization algorithm (Shuffled Complex Evolution,

Duan et al. (1994)), which is recommended for smaller parameter spaces as is the case here (Arsenault et al., 2014).

## 2.2 Case study and hydrometeorological data sets

### 2.2.1 Study area

The study is based on a set of 30 Canadian catchments spread over the Province of Quebec. These catchments' temporal streamflow patterns are primarily influenced by Nivo-pluvial events (snow accumulation, melt, and rainfall dynamics) during





spring and pluvial events during spring and fall. The prevailing climate is humid continental, Dfb, according to the Köppen
classification (Kottek et al., 2006). Land uses are mainly dominated by mixed woods, coniferous forests, and agricultural lands.
Figures 2 and 3 display their location and their hydrological regime, respectively.

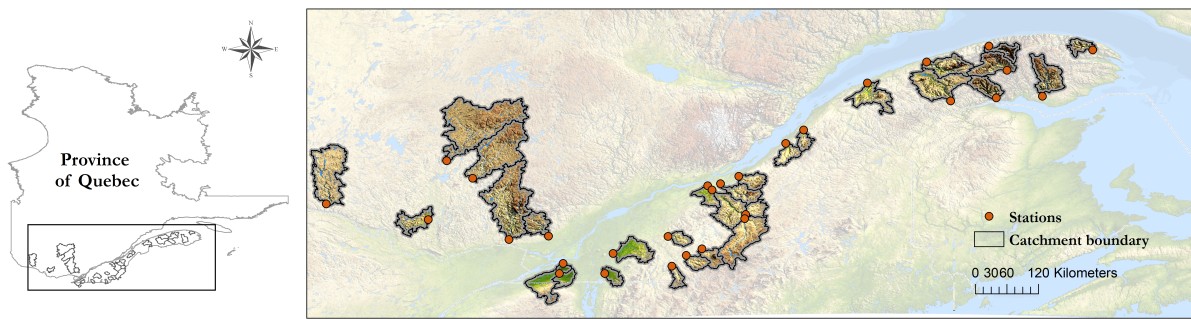

**Figure 2.** Spatial distribution of the 30 studied catchments.

Choosing a sample of catchments with quite similar hydrological regimes may seem limiting, particularly to achieve generalizable results. Nevertheless, the catchments chosen here present differences in their physical features and hydrological

signatures (Table 3).

### 2.2.2  Hydrometeorological observations

The time series of observations extend over 22 years, from January 1995 to December 2016. They were provided by the
"Direction de l'expertise hydrique du Québec" (DEH). They consist of precipitation, minimum and maximum air temperature,
and streamflow series at a three-hour time step. Climatological data stem from station-based measurements interpolated on a

$0.1°$ resolution grid using ordinary kriging. For temperature, kriging is applied at the sea level using an elevation gradient
of -0.005 °C m$^{-1}$. The entire study area is located south of the $50^{th}$ parallel, considered as a region with higher quality
meteorological observations because of the density of the ground-based network (Bergeron, 2016).

Concerning the river discharge series, the DEH's hydrometric stations network records data continuously, every 15 minutes,
and transmits measurements each hour to an integrated collection system where they are subsequently processed and validated.

However, despite constant monitoring and improvements in measurement strategies, these series have missing values during
winter since river icing causes a time-varying redefinition of the flow conditions, resulting in highly unreliable measurements
(Turcotte and Morse, 2016). Accordingly, the winter period (December-Mars) will not be included in the analysis.

In this study, we followed Klemeš (1986) by dividing the available series into two segments: 1997-2007 for calibrating
model parameters and 2008-2016 for computing the goodness of fit. The three previous years of each period allowed for the

spin-up of the models.





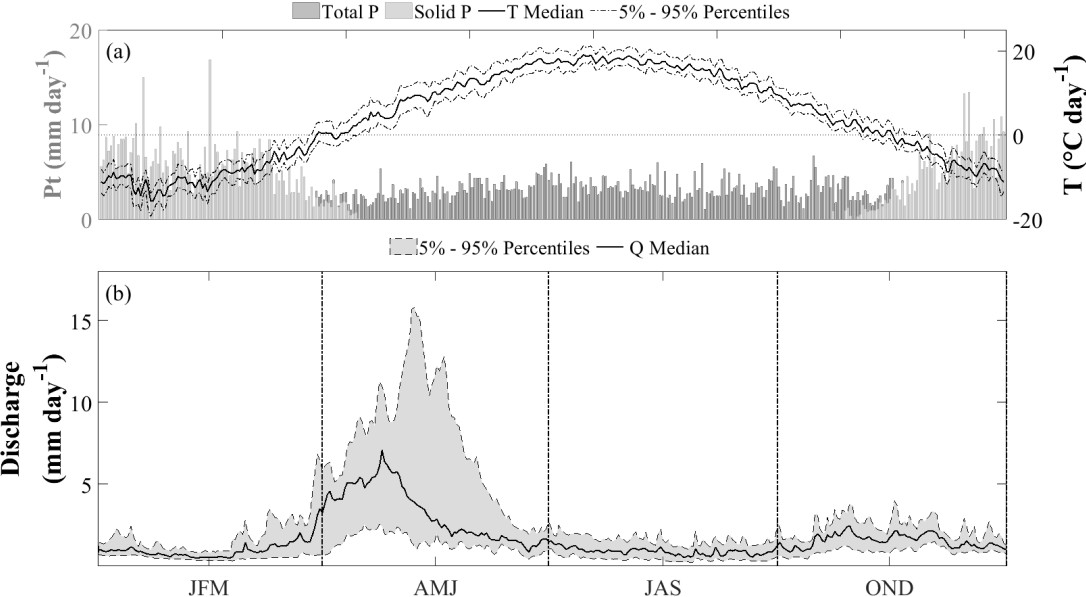

**Figure 3.** Hydrological regime representative of the 30 catchments. (a) The solid black line corresponds to the median of the interannual mean temperature of the catchments; the dotted lines represent the 5 % and 95 % percentiles of interannual mean temperature. Light and dark bars represent the solid and total precipitation, respectively. (b) The solid black line represents the median of the catchments' interannual flows; the gray shaded area the 5 % and 95 % quantiles of interannual flows. Statistics were computed over the observation record available (22 years: 1995-2016).

**Table 3.** Main characteristics of the 30 catchments. Mean annual values and the coefficient of variation were computed over 1995-2016.

| Descriptor (reference) | Abbreviation [unit] | Min. | Med. | Max. |
|---|---|---|---|---|
| Surface | S [km$^2$] | 514 | 1158 | 6768 |
| Mean elevation | Zm [m] | 70 | 362 | 583 |
| Mean annual total precipitation | Ptm [$mm\,y^{-1}$] | 891 | 1013 | 1170 |
| Mean annual solid precipitation (L'hôte et al., 2005) | Psm [$mm\,y^{-1}$] | 379 | 613 | 756 |
| Mean annual evapotranspitation (Oudin et al., 2005) | ETPm [$mm\,y^{-1}$] | 440 | 531 | 626 |
| Mean annual runoff | Qm [$mm\,y^{-1}$] | 403 | 634 | 946 |
| Coeff. of variation (Donnelly et al., 2016) | CV [-] | 132 | 234 | 344 |





## 2.3 Forecast evaluation

A multi-criteria evaluation is applied to measure different facets contributing to the overall quality of the forecasts. We primarily consider scores commonly used in ensemble forecasting to evaluate accuracy, reliability, sharpness, bias, and overall performance (Brown et al., 2010; Anctil and Ramos, 2018). Verification was conditioned on lead time and catchment size over 2011-2016. To highlight the sensitivity of the results to catchment size, we defined three catchment groups: smaller (< 800 km$^2$: 11 catchments), medium (between 800 km$^2$ and 3,000 km$^2$: 10 catchments), and larger (>= 3,000 km$^2$: 9 catchments).

To increase the readability of the text, the equations for each of the selected metrics have been placed in an appendix. Figure 4 proposes a graphical explanation for some of them.

### 2.3.1 Evaluation criteria

The relative bias (BIAS) is used to measure the overall unconditional bias (systematic errors) of the forecasts (Anctil and Ramos, 2018). Mathematically, it is defined as the ratio between the mean of the ensemble average and the mean observation. The BIAS is sensitive to the direction of errors: values higher (lower) than 1 indicate an overall overestimation (underestimation) of the observed values.

The continuous ranked probability score (CRPS) is a common metric to measure the overall performance of forecasts (Fig. 4a). It represents the quadratic distance between the cumulative distribution function (CDF) of the forecasts and the empirical CDF of the observations (Hersbach, 2000). The CRPS shares the same unit as the predicted variable. A value of 0 indicates a perfect forecast, and there is no upper bound. As the CRPS assesses the forecast for a single time step, the MCRPS is defined as the average CRPS over the entire evaluation period. We estimate the CRPS from the empirical CDF of forecasts.

Reliability is the alignment between the forecast probabilities and the frequency of observations. It describes the conditional bias related to the forecasts. In this study, it was evaluated using the spread-skill plot (SSP, Fortin et al., 2014) and the reliability diagram (Wilks, 2011). The SSP evaluates the ability of the ensemble spread (the square root of average ensemble variance) to represent the forecast error, expressed as the Root Mean Square Error (RMSE, a measure of skill). In a reliable system, the ensemble correctly predicts the forecast error (i.e., the spread and the RMSE should match). However, the system is underdispersed (overdispersed) when the RMSE is superior (inferior) to the spread. The reliability is thus somehow decomposed into an accuracy error part and a spread component.

The reliability diagram is a graphical verification tool that plots forecasts probabilities against observed event frequencies (Fig. 4b). The range of forecast probabilities is divided into $K$ bins according to the forecast probability (horizontal axis). The sample size in each bin is often included as a histogram. A perfectly reliable system is represented by a 1:1 line, which means that the probability of the forecast is equal to the frequency of the event. If the curve falls above this line, it indicates an overdispersion (underforecasting: the observations are too often in the low end of the ensemble spread) and underdispersion (overforecasting), otherwise. Reliability diagrams depend on events definition (Pt $\geq$ a threshold). Data stratification when performing conditional verification (e.g., thresholds or seasons) makes the time series shorter, which may lead to biases and possible inaccurate conclusions concerning the performance of the implemented techniques (Lerch et al., 2017; Bellier et al.,





2017b). To deal with this drawback, while assessing precipitation forecasts, some authors choose to study three thresholds for

exceeding low, intermediate, and high values of precipitation (Scheuerer and Hamill, 2018). However, as noted by Pappenberger et al. (2008), thresholds used in precipitation forecast evaluation are not necessarily linked to the dominant hydrological processes. Accordingly, we decided to use the reliability diagram to evaluate precipitation for thresholds of different exceedance probabilities (EP), namely 0.05, 0.5, 0.75, and 0.95. To compare the reliability score of precipitation with the reliability score of streamflow forecasts (and for practical purposes and simplification), we use the mean absolute error from the reliability dia-

gram (MAE$_{rd}$, Fig. 4c). In this case, the MAE$_{rd}$ measures the distance between the predicted reliability curve and the diagonal (perfect reliability), allowing the reliability diagram to be seen as a numerical verification tool.

To measure the degree of variability of the forecasts, or the sharpness of the ensemble forecasts, we use the 90 percent interquartile range (IQR). It is defined as the difference between the $95^{th}$ and the $5^{th}$ percentiles of the forecast distribution (Fig. 4d). The narrower the IQR, the sharper the ensemble. As the sharpness is a property of the unconditional distribution of

forecasts only (sharp forecasts are not necessarily accurate or reliable; sharp forecasts are accurate if they are also reliable), we use this attribute as a complement to the reliability (i.e., given two reliable systems, sharper is better) (Gneiting et al., 2007). The frequency of forecasts shown in the reliability histogram (Fig. 4b) gives information on sample size and sharpness as well. A sharp forecast tends to predict probabilities near 0 or 1.

We use the volumetric efficiency (VE, Criss and Winston, 2008) to evaluate and compare the performance of the seven

hydrological models. The VE represents the fractional volumetric difference between the simulated and observed streamflows. The EV ranges from 0 to 1. A perfect value of 1 indicates that the volume of water predicted by the model matches the observed volume. This criterion gives the same weight to any flow range (e.g., slow recession and rapid rising flow), relaxing the constraint of model residuals heteroscedasticity.

The modified Kling-Gupta efficiency (KGEm, Kling et al., 2012) was used as objective function to identify the optimal set

of parameters, but also as criteria to evaluate the quality of model outputs in the calibration and validation of the hydrological models. It assesses the shape, timing, water balance, and variability of discharge time series. KGEm values range between $-\infty$ and 1 (positively oriented), the latter being the perfect value.

### 2.3.2  Skill scores

Skill scores (SS) are used to evaluate the performance of a forecast system against the performance of a reference forecast. The

criteria described in Sect. 2.3.1 can be transformed into a SS by using the relationship described in the Eq. (6):

$$SS = 1 - \frac{Score^{Syst}}{Score^{Ref}} \tag{6}$$

where $SScore^{Syst}$ and $SScore^{Ref}$ are the scores of the forecasting system and the reference, respectively. The SS values range from $-\infty$ to 1. If SS is superior (inferior) to zero, the forecast performs better (worst) than the reference. When it is equal to zero, both systems have the same performance or skill.





Since our goal is to determine the value added by a precipitation post-processor in the quantification of streamflow forecasting uncertainty, we use the raw forecasts as benchmark (Pappenberger et al., 2015).

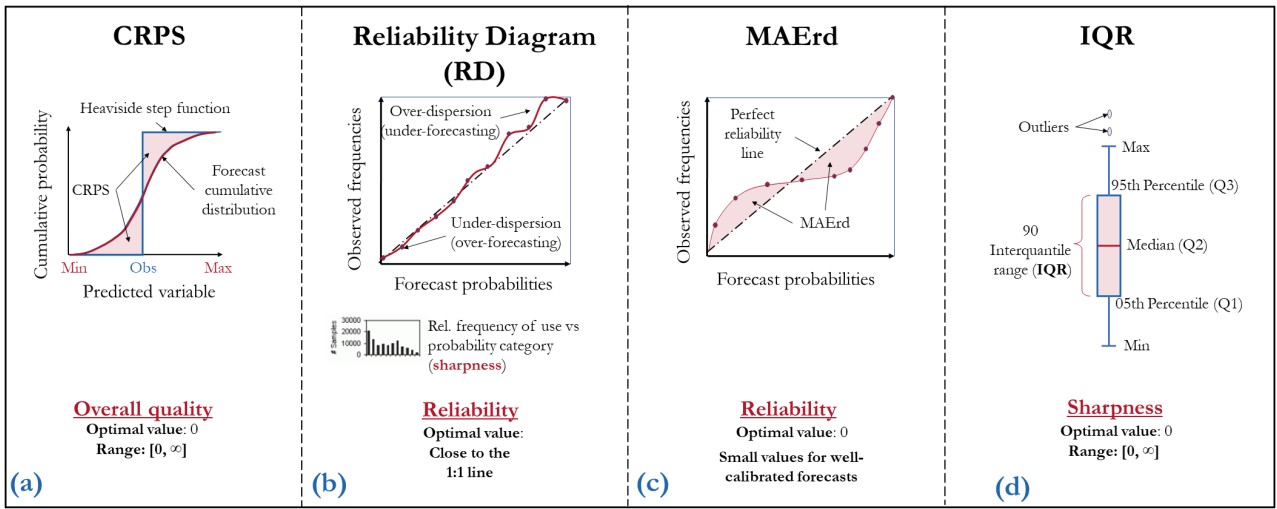

**Figure 4.** Graphical representation of the CRPS, Reliability Diagram, MAE of the Reliability Diagram (MAE$_{rd}$), and IQR forecast evaluation criteria.

## 3   Results

Results are presented as follow: the performance of the seven hydrological models (Sect. 3.1), the performance of the raw ensemble hydrometeorological forecasts (Sect. 3.2), the performance of the post-processed hydrometeorological forecasts (Sect.

3.3), and the performance of the precipitation post-processor conditioned to catchment size (Sect. 3.4).

### 3.1   Performance of the hydrological models

An exhaustive evaluation of the quality of each hydrological model is not the objective of this study. However, it is important to provide information on how the models represent the precipitation-streamflow response of the catchments. The evaluation is based on volumetric efficiency (VE), KGEm, and relative bias (BIAS). Figure 5 shows the seven hydrological models'

performance in calibration (1997-2007) and validation (2008-2016). The boxplots correspond to calibration (red), validation without DA (light blue), and validation with DA (dark blue). Each boxplot summarizes the distribution (minimum, quantiles 0.25, 0.5, and 0.75, and maximum) of the 30 catchments. As Fig. 5 shows, no model is systematically better than the others for the different criteria and over the catchments. Overall, models present a better performance in calibration, as generally expected. When using EnKF DA, some models have slightly better or similar VE and KGEm values in validation (e.g., M2,

M3, M5, and M6), although the EnKF DA does not display a uniform influence on the models. Models M1, M4, and M7





exhibit lower transposability; that is, the parameters estimated in calibration are less suitable for the validation period (a drop in performance in validation shows this).

In terms of EV, the fraction of simulated and observed volumes differ on average by 3 % in calibration and 18 % in validation, but improves to 8 % when activating the EnKF DA in validation. The BIAS reveals that this difference is usually an overestimation and confirms the effectiveness of the EnKF, especially to deal with transposability issues between calibration and validation. We obtained an average KGEm of 0.83 in calibration, 0.64 in validation without EnKF, and 0.82 in validation with EnKF, over the 30 catchments.

Figure 6 presents the interannual hydrographs of the catchments with the best and the worst performance spanning over the calibration and validation periods. The hydrographs reveal that the overestimation occurs mostly in spring, during snowmelt. Overall, the ensemble of hydrological models reproduces the hydrographs well.

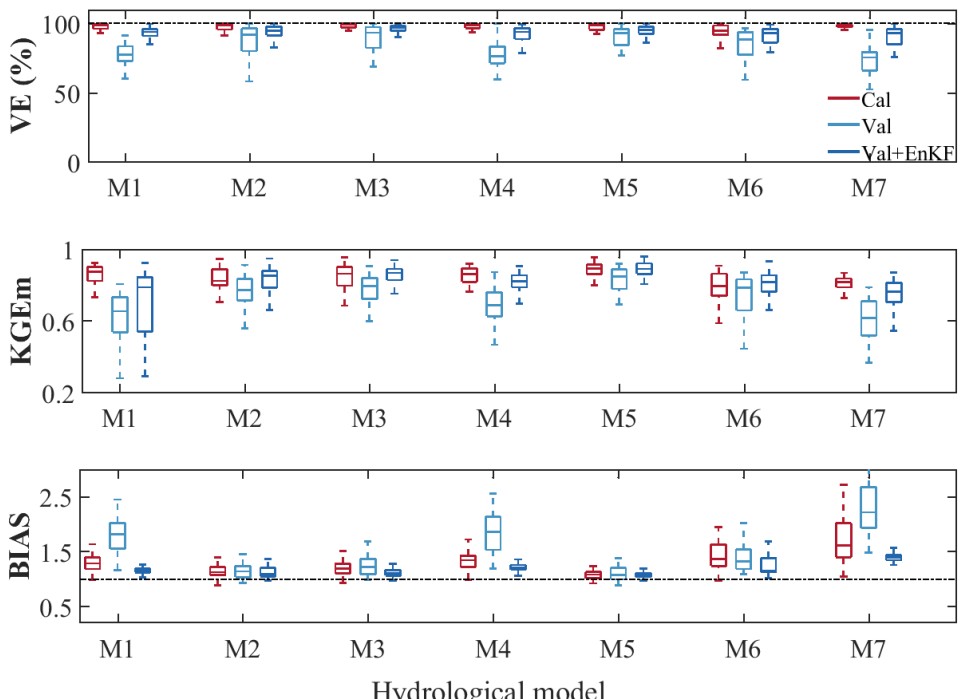

**Figure 5.** Performance of the seven hydrological models in terms of volumetric efficiency (VE), KGEm, and relative bias (BIAS) in calibration (1997-2007; red), validation (2008-2016) without EnKF DA (light blue), and with EnKF DA (dark blue). Boxplots represent the distribution of the scores over 30 catchments. The scores exclude wintertime (December-March).





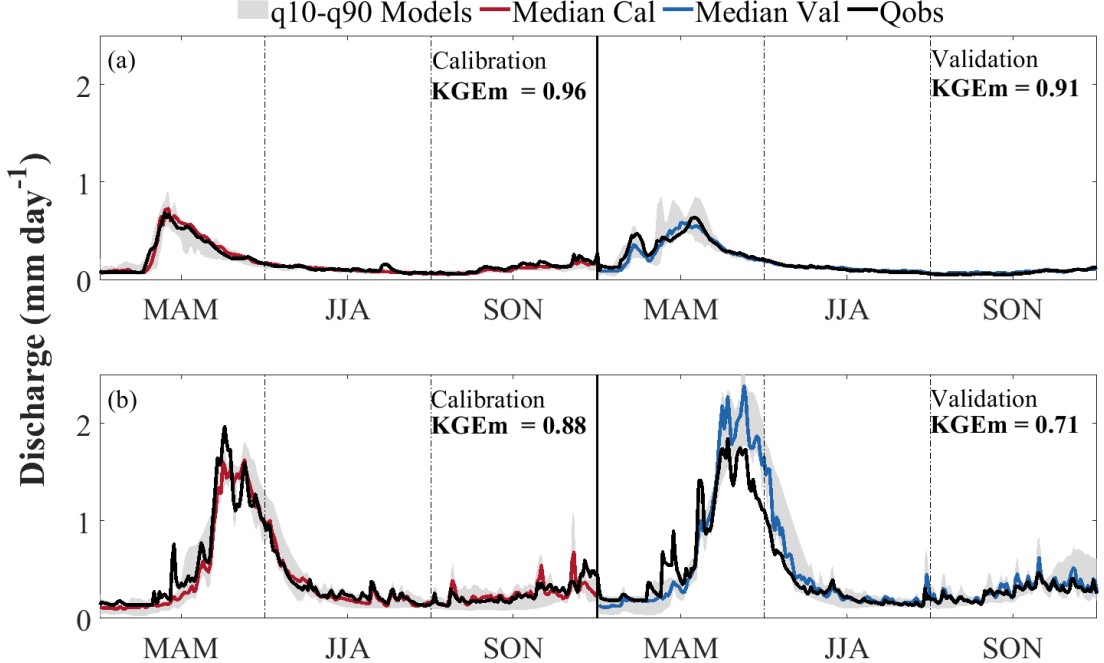

**Figure 6.** Simulated and observed (black) interannual streamflow of the catchments with the best (a) and the worst (b) performance in calibration (1997-2007; red) and validation (2008-2016; blue). The grey shaded area represents the variability (80 % interval) of the seven hydrological models. MAM: March-April-May; JJA: June-July-August; SON: September-October-November.

## 3.2 Performance of the raw forecasts

This section assesses the quality of the ECMWF precipitation forecasts (average precipitation over the catchments) and the quality of streamflow ensemble forecasts based on raw precipitation forecasts. Values presented are average daily scores over the evaluation period (2011-2016).

### 3.2.1 Precipitation

Figure 7 presents the distribution of the scores of precipitation forecast quality as a function of lead time. Each boxplot combines values from all catchments. As expected, the quality of the forecasts generally decreases with increasing lead times. BIAS values greater than one indicate that the precipitation forecasts overestimate the observations. The MCRPS shows that the overall forecast quality decreases with lead time while reliability improves, as revealed by the Reliability Diagram Mean

Absolute Error ($MAE_{rd}$). This is a general characteristic of weather forecasting systems, as the dispersion of members increases with the forecast horizon to capture increased forecast errors. Reliability improvement is reflected in the BIAS, where a slight decrease in the overestimation is observed from day five onwards. The typical trade-off between reliability and sharpness is





also illustrated (the MAE$_{rd}$ vs. the Interquartile Range (IQR)). IQR has an opposite behavior to the MAE$_{rd}$, indicating a less sharp forecast with increased lead time.

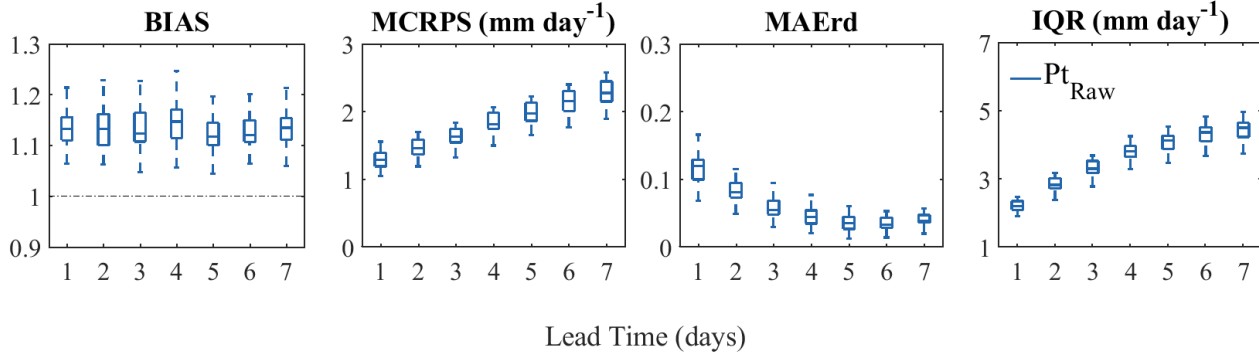

**Figure 7.** BIAS, MCRPS, MAE$_{rd}$ and IQR of raw daily catchment-based precipitation forecasts as a function of lead time for all catchments and the evaluation period (2011-2016). Boxplots represent the distribution of the scores over 30 catchments.

### 3.2.2 Streamflow


The performance of the hydrological forecasting systems is illustrated in Fig. 8. Each score is presented for lead times of 1, 3, and 6 days. As summarized in Table 1, systems are ordered from the simplest (A: one source of uncertainty) to the most complex (D: three sources of uncertainty).

BIAS indicates that systems B and C present the highest and lowest performance, respectively. In fact, the systems that

benefit from EnKF (i.e., B and D) show better performance, confirming the positive EnKF DA contribution as seen in the calibration and validation evaluation (Sect. 3.1), especially at the first lead times. As the EnKF DA updates model states only once (at the time the forecast is issued), its effects fade out over time. This explains why systems A and B tend to behave similarly as lead time increases. System C, which exploits multiple models, overestimates the observations in most catchments, especially at the first lead time. This behavior is inherited from the hydrological models, as shown in Figs. 5 and 6. The BIAS

evaluation also reveals that catchment diversity is one factor explaining differences in performance. As lead time increases, forecasts tend to underestimate the observations for some catchments.

As in the BIAS, systems that quantify the uncertainty of the initial conditions (B and D) are the ones that present the best MCRPS. The improvement brought by these systems could be attributed to the fact that these are the two systems with the largest number of members. Studies have shown that sample size influences the computation of some criteria, such as the CRPS

(Ferro et al., 2008). However, Fig. 8 shows that the effect of the quantification of uncertainty sources is more critical than the ensemble size since systems B, C, and D present a similar range of score when the contribution of EnKF is minimal (i.e., at day 6). These results are in agreement with those found by Thiboult et al. (2016) and Bourgin et al. (2014), who suggested





that short-range forecasts benefit most from data assimilation. From Fig. 8 we can also see that system A presents the most unfavorable scenario, which is expected since it only carries meteorological forcing uncertainty with it, and the accuracy of weather forecasts tends to decrease with lead time (Fig. 7).

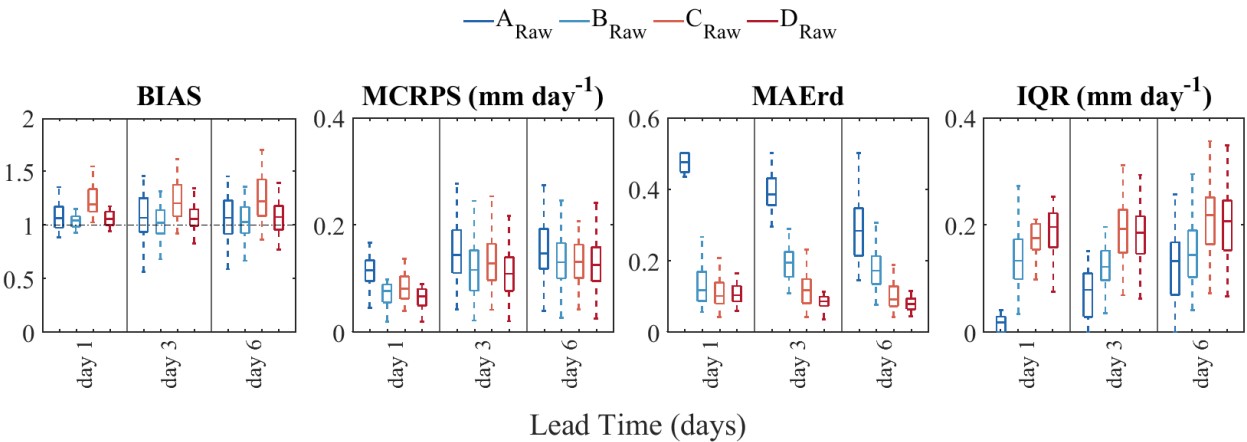

**Figure 8.** BIAS, MCRPS, MAE$_{rd}$ and IQR of streamflow forecasts based on raw precipitation forecasts for four hydrological forecasting systems and lead times of 1, 3, and 6 days over the evaluation period (2011-2016). Boxplots represent the distribution of the scores over 30 catchments. The systems differ in the quantification of uncertainty: (A) forcing, (B) forcing and data assimilation, (C) forcing and model structure, and (D) forcing, model structure, and data assimilation (see Table 1).

The MAE$_{rd}$ and IQR, together, allow us to evaluate the contribution of each tool for uncertainty quantification in terms of their ability to capture the total uncertainty over time. The MAE$_{rd}$ shows that system A follows a pattern similar to the weather forecasts (Fig. 7), becoming more reliable with increasing lead times but less sharp (higher values of IQR). In general, when a system has an underdispersion, it is sharper but unreliable. System B loses reliability at day 3 because the EnKF effects fade out over time. However, it becomes slightly more reliable on day 6 because of the spread of the precipitation forecasts. This is also the case with system C, which also benefits from the meteorological ensemble, but, unlike system B, its performance remains almost constant over time thanks to the hydrological multimodel. System D is less reliable on day 1. According to Thiboult et al. (2016), the combination of EnKF and the multimodel ensemble causes an overdispersion since the EnKF indirectly quantifies uncertainty from the hydrological model structure and its parameters when performing DA with the estimation of initial conditions uncertainty. Nevertheless, forecast overdispersion is reduced as the EnKF effects vanish with lead time, and the system becomes more reliable on day 6. Although the EnKF loses its effectiveness over time, the difference between systems C and D on days 3 and 6 reveals that its contribution to forecasts performance is still important at longer lead times.





## 3.3 Performance of the post-processed hydrometeorological forecasts

This section compares the quality of the raw and the post-processed precipitation forecasts and assesses the contribution of the precipitation post-processor to each hydrological forecasting system. Scores are presented for days 1, 3, and 6 in boxplots representing all catchments.

### 3.3.1 Performance of corrected precipitation forecast

Figure 9 presents BIAS, MCRPS, MAE$_{rd}$ and IQR scores of the raw and post-processed precipitation forecasts. As illustrated,
the CSGD post-processor substantially reduces the relative bias of the meteorological forecasts since day 1, and its effectiveness is maintained over time and for all catchments. When considering MCRPS, the performance of the CSGD post-processor decreases when increasing lead times. This result is expected as the predictors use information from the raw forecasts (Scheuerer and Hamill, 2015), which also decrease in MCRPS quality with lead time (Fig. 7). In terms of MAE$_{rd}$, the post-processed and raw forecasts have an inverse behavior: while reliability increases rapidly with lead time for raw forecasts, it slightly decreases
for post-processed forecasts. We note that, at day 6, raw forecasts are more reliable, on median values over the catchments, while also being sharper than post-processed forecasts.

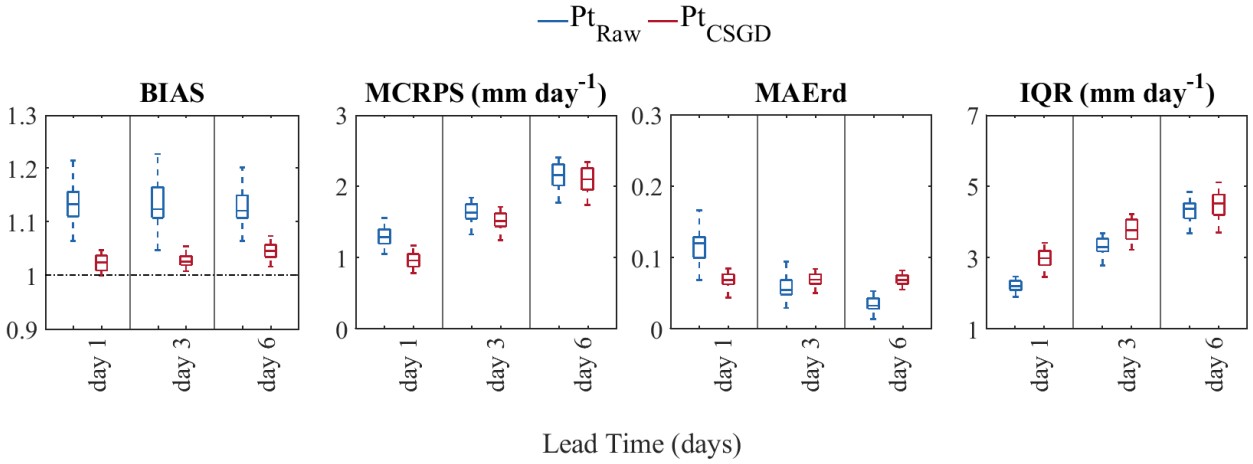

**Figure 9.** BIAS, MCRPS, MAE$_{rd}$ and IQR of raw (blue) and post-processed (red) daily catchment-based precipitation forecasts for lead times of 1, 3, and 6 days over the evaluation period (2011-2016). Boxplots represent the distribution of the scores over 30 catchments.

Figure 10 shows the reliability diagrams for the probability of raw and post-processed precipitation forecasts exceeding the 0.05, 0.5, 0.75, and 0.95 quantiles for lead times 1, 3, and 6 days. The confidence bounds shown in the curves result from a bootstrap with 1000 random samples representing the 90 % confidence interval. The curves represent the median curve of the
30 catchments. The inset histograms depict the frequency with which each probability was issued. The plot confirms the results in Fig. 9. In general, the performance of the CSDG post-processor in terms of reliability decreases with increasing lead times





and precipitation thresholds. The CSGD post-processed precipitation forecasts are already reliable at short lead times (day 1) except for the exceedance probability (EP) event of 0.05, which in fact corresponds closely to the probability threshold of zero precipitation. For this threshold, the CSGD post-processor tends to generate forecasts that underestimate the observations,
although this trend decreases as lead time increases. This can be attributed to the fact that the CSGD post-processor retains the climatological shift parameter ($\delta=\delta_{cl}$) that tends to produce a bias in $POP_f$ estimates (Ghazvinian et al., 2020).

Regarding the raw precipitation forecasts, they tend to underforecast the low probabilities and overforecast the high ones. Similarly, as shown in Fig. 9, raw precipitation reliability increases with lead time except for large precipitation amounts. However, in both cases, the more reliable the forecast, the flatter the histogram. This indicates that the forecasts predict all
probability ranges with the same frequency, and therefore the system is not sharp (a perfectly sharp system populates only 0 % and 100 %).

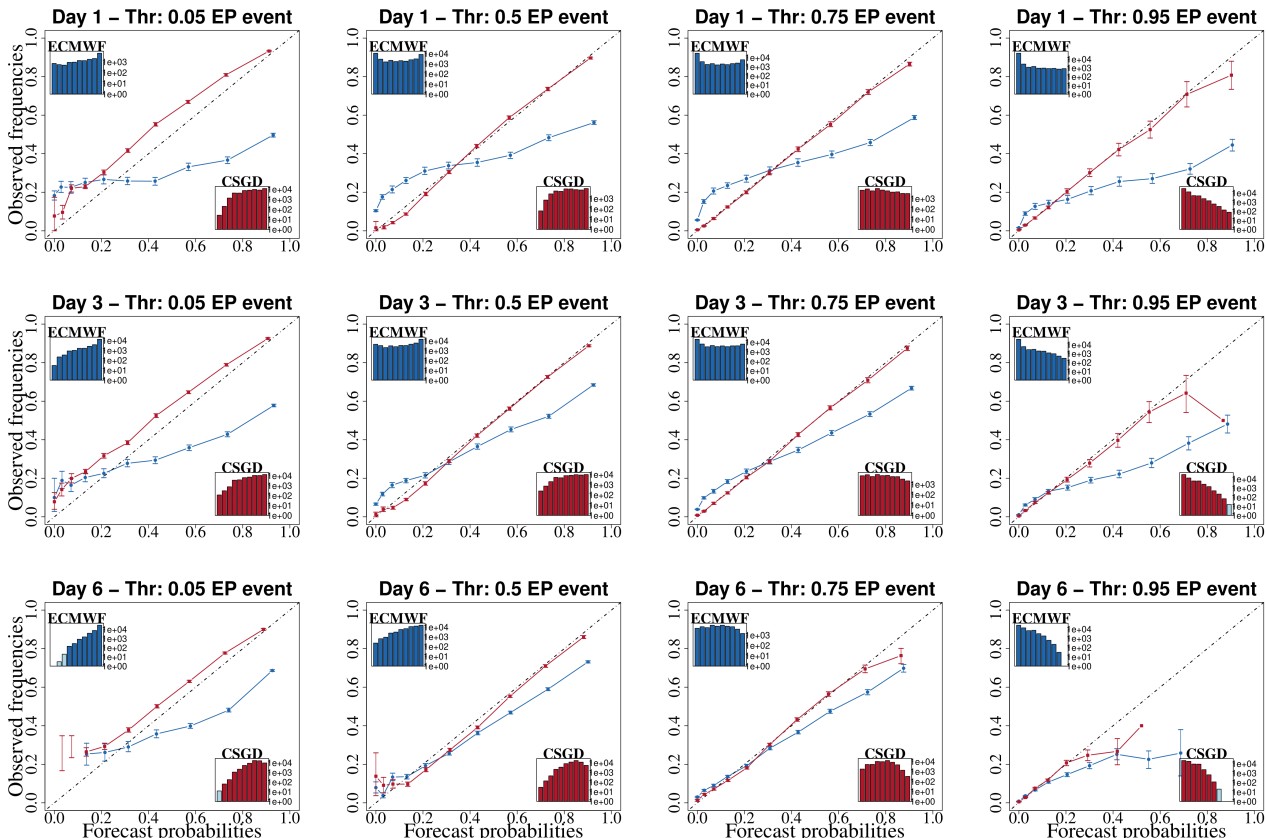

**Figure 10.** Reliability diagrams of raw (blue) and CSDG post-processed (red) precipitation forecasts for lead times of 1, 3, and 6 days and different exceedance probability (EP) thresholds (> 0.05, 0.5, 0.75, 0.95 quantile of observations), calculated over the evaluation period (2011-2016). The lines correspond to the median curve of all the 30 catchments. The bars indicate 90 % confidence intervals of observed frequencies from bootstrap resampling. The inset histograms depict the frequencies with which the category was issued.





### 3.3.2 Interaction of the precipitation post-processor with the hydrological forecasting systems

Figures 11 to 14 illustrate the main interactions of the precipitation post-processor with the hydrological forecasting systems. For instance, the improvements brought by the precipitation post-processor in terms of BIAS (Fig. 11) are only reflected in the streamflow forecasts in the farthest lead times, when streamflow forecasts based on post-processed precipitation forecasts display better performance than streamflow forecasts based on raw precipitation forecasts. We also observe that post-processing precipitation forecasts have a much higher impact on the quality of precipitation forecasts (Fig. 9) than on the quality of streamflow forecasts, as evaluated by the BIAS score (Fig. 11). On day 1, the CSGD post-processor does not affect streamflow forecasts, except for system A, which is a system that depends exclusively on the ability of the ensemble precipitation forecasts to quantify forecast uncertainty. On days 3 and 6, the streamflow forecasts present, on average, better performance when they are based on post-processed precipitation forecasts. Additionally, the increased bias on days 3 and 6 in system A reveals that at these lead times, forcing uncertainty does not represent the dominant source of uncertainty for some catchments. This implies that a simple chain with a precipitation post-processor may be insufficient to systematically provide unbiased hydrological forecasts. At these lead times, in terms of BIAS, the performance of raw forecasts in systems B and D are generally better than the performance of post-processed forecasts in system A. System C, which has the lowest performance, is also improved by the precipitation post-processor. However, this improvement still shows the worst performance compared to the other systems based on raw precipitation forecasts.

In terms of overall quality (MCRPS), the streamflow forecasts based on post-processed precipitation forecasts perform better for systems A and B, but only at the shorter lead times (Fig. 12). In the longer lead times, the effect of the CSDG post-processor is much less important. Systems C and D, on the other hand, appear to be unaffected by the post-processing of precipitation forecasts at the short lead times and be slightly improved by the post-processor with increasing time. We also observe that the performance of raw forecasts in all systems, notably in systems B and D, is generally better than post-processed forecasts in system A. Differences are higher at shorter lead times and tend to decrease at longer lead times. This indicates that benefits are brought by quantifying more sources of uncertainty, especially at shorter lead times, instead of just relying on forcing uncertainty quantification through post-processed ensemble precipitation forecasts to enhance the overall performance of ensemble streamflow forecasting systems.

In terms of reliability (Fig. 13), systems that use a single hydrological model (A and B) are the ones that benefit the most from post-processing precipitation forecasts. When a multimodel approach is used (C and D), the system becomes more robust, and differences in streamflow forecast quality from using or not a precipitation post-processor are small. On days 3 and 6, the contribution of the CSGD precipitation post-processor is more important for system B. It seems that the CSGD overdispersion (Fig. 10) compensates for the loss of dispersion from the use of EnKF DA. It is interesting to note that, for systems B, C, and D, streamflow forecasts based on raw precipitation forecasts are always much better than streamflow forecasts based on post-processed precipitation forecasts in system A. similarly to the bias and the overall performance, this indicates that forcing uncertainty only is not enough to deliver reliable streamflow forecasts, and quantifying other sources of hydrological uncertainty can be more efficient than only post-processing precipitation forecasts.



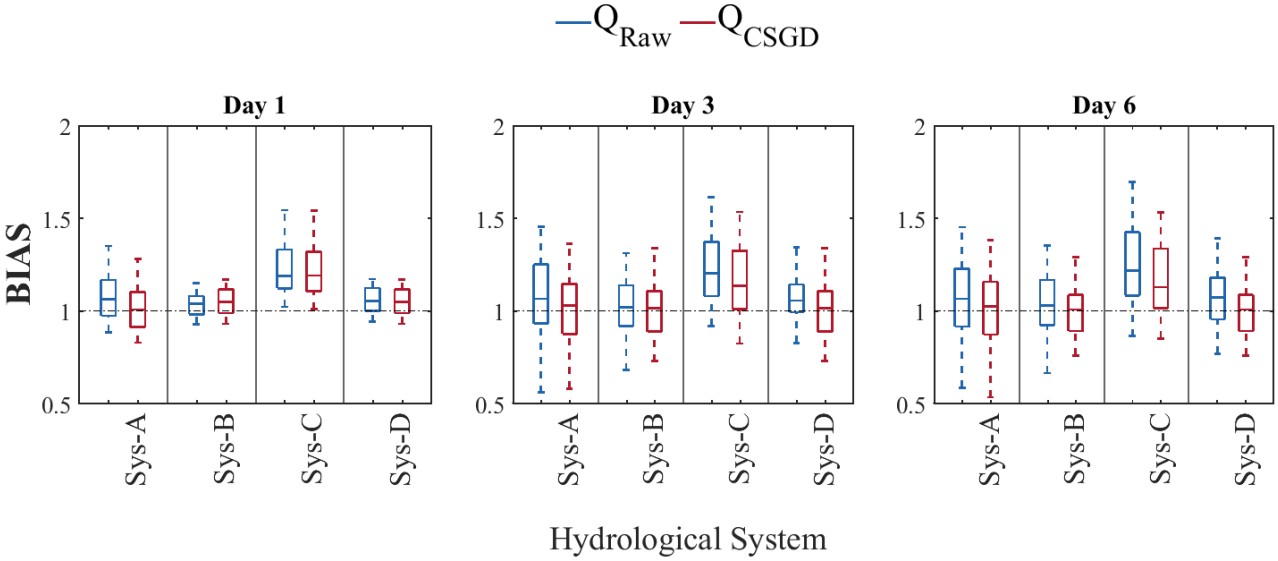

**Figure 11.** Relative bias (BIAS) of the ensemble streamflow forecasts of the four hydrological prediction systems for lead times 1, 3, and 6 days when considering raw (blue) and post-processed (red) precipitation forecasts. Boxplots represent the score variability over the 30 catchments.

For the IQR (Fig. 14), we again see a trade-off with reliability. The increased dispersion that contributes to the reliability of the systems makes the forecasts less sharp. For example, system B, which shows the greatest benefit from the precipitation post-processing in terms of reliability, is also the one that is less sharp when compared to its raw forecast counterpart. Finally, although post-processed systems C and D display similar reliability scores (Fig. 13), the more complex system does not display

sharper forecasts (Fig. 14). The gain in system's complexity does not translate into an important gain in reliability/sharpness of the forecasts.

### 3.4   Effect of catchment size

Figure 15 shows spread-skill plots (SSP) of the median results of the catchments analyzed according to three groups (smaller, medium, and larger) and for the seven days of lead time. The solid lines correspond to the RMSE and the dashed lines to

the spread. The first column corresponds to the SSP of precipitation forecasts and the remaining to the SSP of streamflow forecasts issued by the four hydrological prediction systems. In general, the performance of precipitation and streamflow forecasts increases with catchment size, as RMSE values are lower for larger catchments and closer to the spread values. This is particularly observed in the streamflow forecasts and may be related to the use of lumped hydrological models, which tend to smooth errors over larger modeling areas (Andréassian et al., 2004).



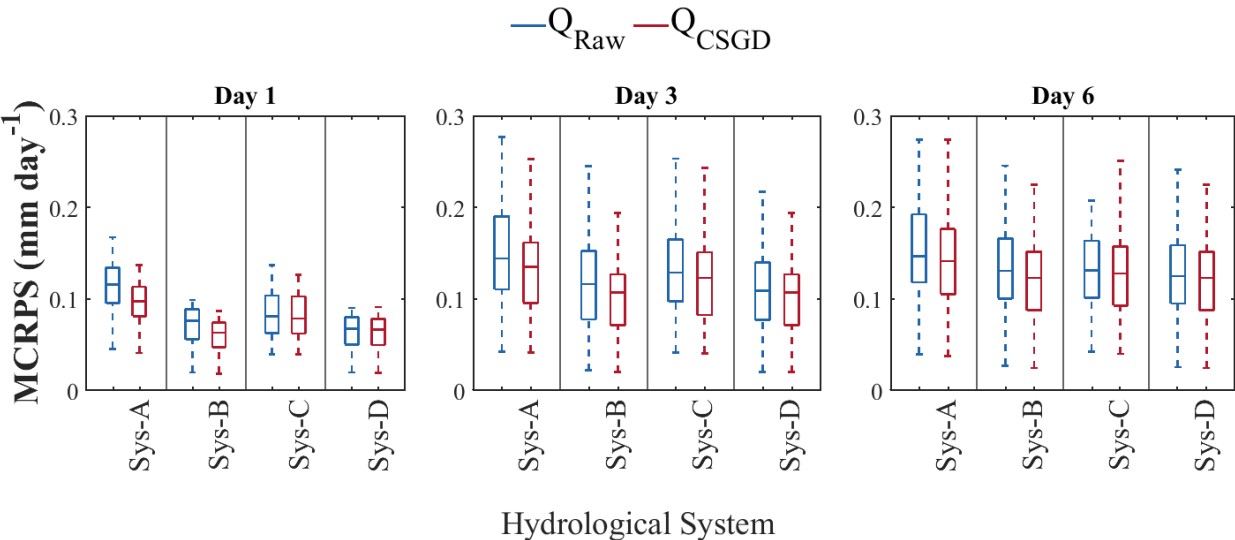

**Figure 12.** Same as Figure 11 but for MCRPS.

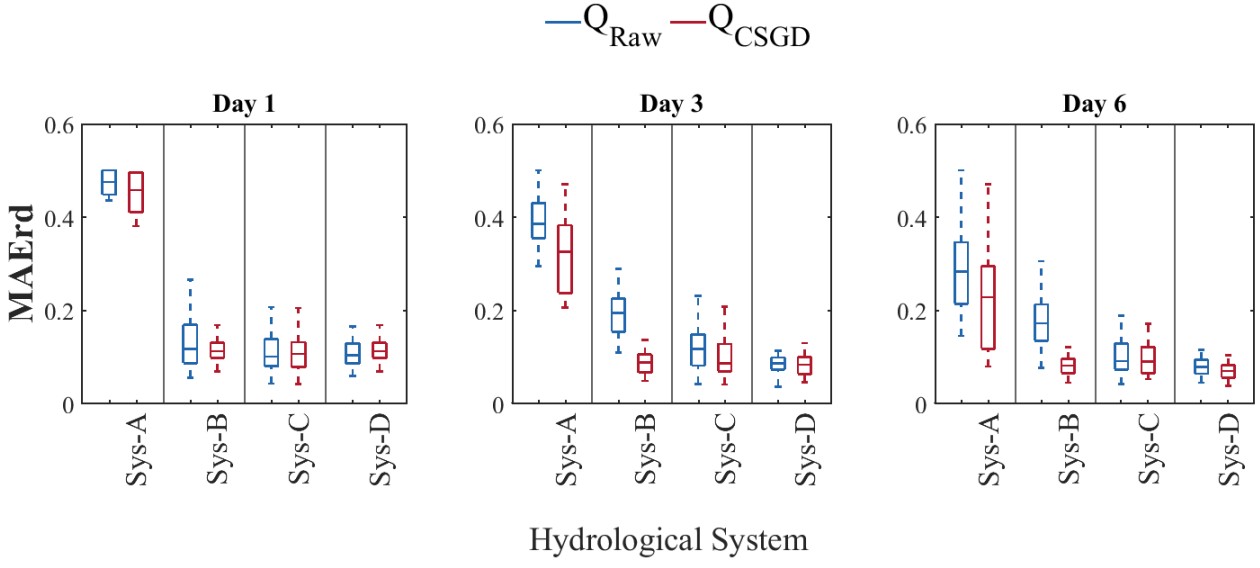

**Figure 13.** Same as Figure 11 but for MAEdf.

When considering the group of smaller catchments, we can see that the RMSE and the spread of the post-processed precipitation forecasts tend to converge when moving from day 1 to day 7, indicating a gain in reliability of the system over time.





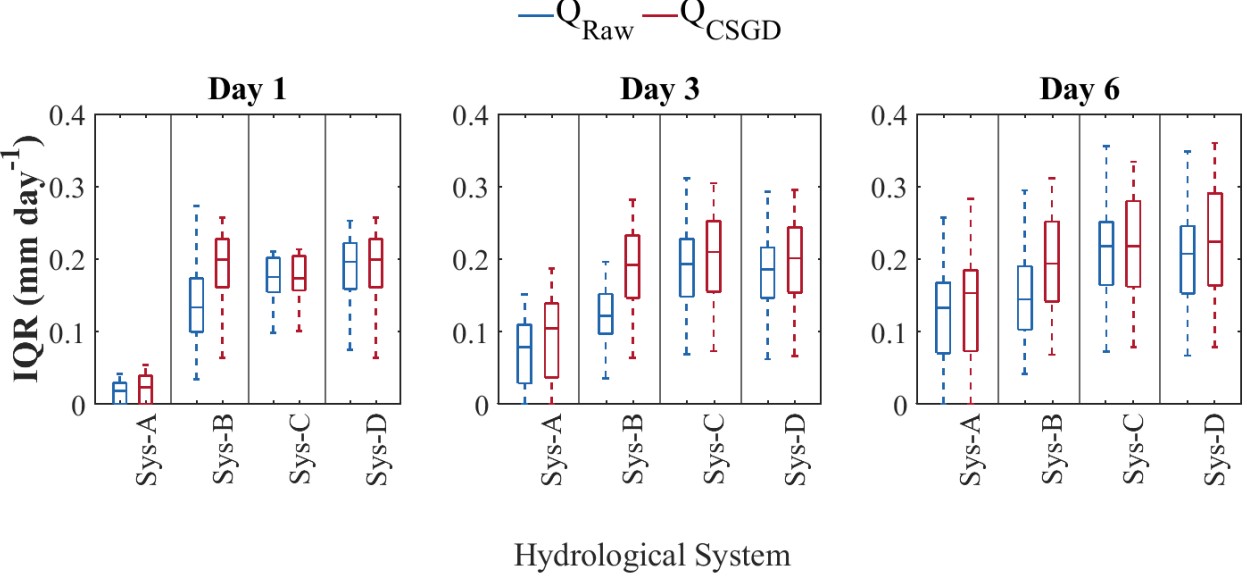

**Figure 14.** Same as Figure 11 but for IQR.

When evaluating the streamflow forecasts, we can see that the contribution of the precipitation post-processor mostly translates into more dispersion (red dotted curves are above blue dotted curves in Fig. 15, while RMSE curves remain superposed). The decrease in error is almost nil for systems A and B and very small for systems C and D in the smaller catchments group. These

findings confirm that large improvements in precipitation forecasts do not necessarily lead to improvements in streamflow forecasts.

    The evolution of the spread of the raw forecasts of this group of smaller catchments illustrates the contribution of each of the tools used to quantify forecast uncertainty in the systems. For example, in system A, the spread is narrow in the first days of forecast and increases with lead time, which is characteristic of and inherited from the meteorological forecasts. For system

B, we see the contribution of EnKF DA and the decrease of spread until day 4 (dashed blue line), when the contribution from the spread of the precipitation ensemble forecast becomes dominant over the reduction of uncertainty from the DA, and the hydrological forecasts start to spread again. In system C, the spread increases with regard to the spread of systems A and B, although its values remain close to the spread values in system D. This shows that the multimodel is the source that contributes the most to increase the dispersion of the ensemble streamflow forecasts in the smaller catchments. The results for system

D show a peak of high spread at day 1. This overdispersion is generated by combining the multimodel and the EnKF DA. However, it does not seem to be impacted by the increase in spread introduced by the post-processor of precipitation forecasts (Fig. 9, IQR).

    For the group of medium-sized catchments, the post-processed ensemble precipitation forecasts are reliable until day three (spread and RMSE red curves for Pt aligned in Fig. 15, central panel), and then they display an overdispersion until day 7. For



longer lead times, it is the raw forecasts that display reliability (spread and RMSE blue curves for Pt aligned in Fig. 15, central panel, after day 4). This confirms the results illustrated in Figs. 9 and 10: 1) the post-processed precipitation forecasts errors are lower than the raw forecasts over time, and 2) reliability increases with lead times for raw forecasts and decreases for post-processed forecasts. Contrary to smaller catchments, when evaluating the streamflow forecasts in this group, the precipitation post-processor improves the hydrological forecasts in terms of error correction rather than in terms of spread. The greatest

contribution of the CSGD precipitation post-processor in catchments of medium size is made in systems A and C. The RMSE and spread (red) curves tend to get closer to each other for these systems. In system B, the contribution is minimal and only in terms of spread up to the first four days of lead time. In system D, the application of precipitation post-processing does not lead to reliable streamflow forecasts: the RMSE and spread curves in red in Fig. 15 are not aligned, while the blue curves (for raw forecasts) are for lead times greater than 3 days.

540        The effect of the precipitation post-processor in streamflow forecasts for the larger catchments group is practically negligible. Red and blue curves are superposed (Fig. 15, bottom panel).

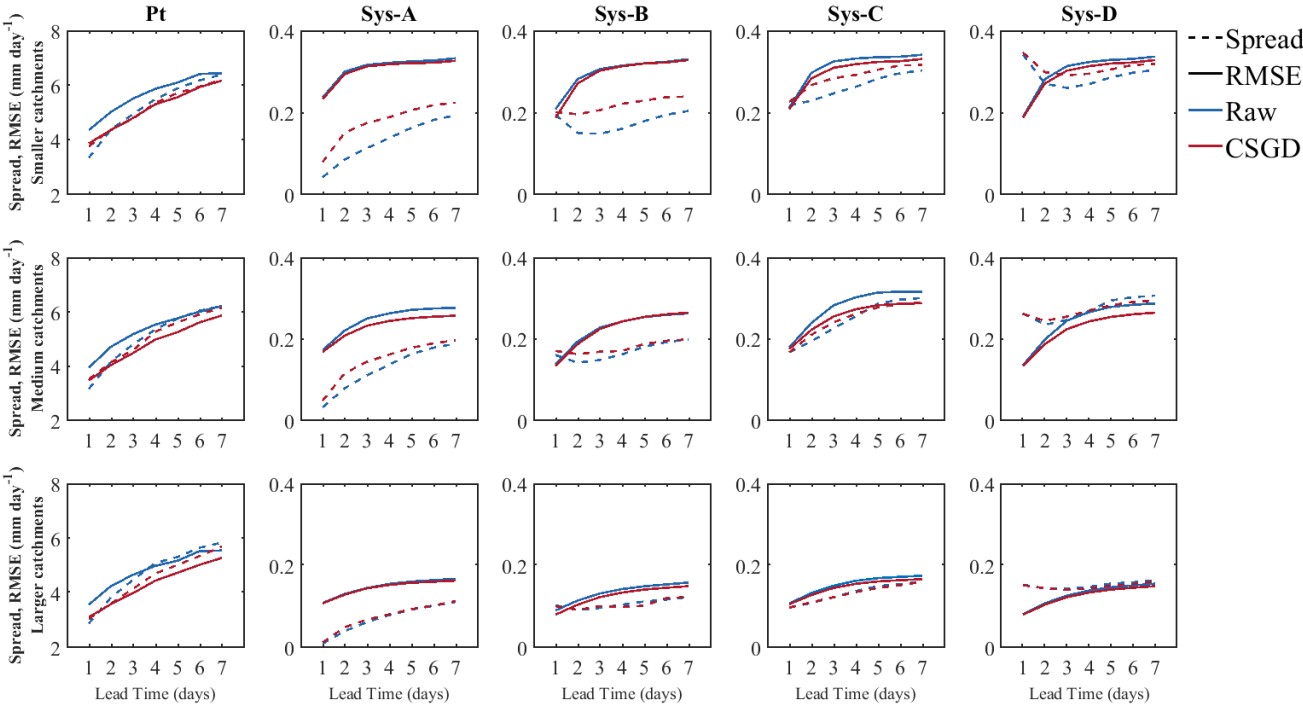

**Figure 15.** Spread-skill plots (SSPs) for raw (blue) and precipitation post-processed (red) forecasts for precipitation (Pt) and streamflow forecasts as a function of forecast lead time. The first column represents the SSP for the raw and post-processed precipitation forecasts. The remaining columns represent the four streamflow forecasting systems (A, B, C, and D). In top: the smaller catchments group. In the middle: the medium catchments group. In bottom: the larger catchments group.





### 3.4.1 Gain in streamflow forecasts from CSGD correction

Figure 16 presents the MCRPS skill scores of precipitation forecasts (Pt-MCRPS SS) against the skill scores of streamflow forecasts (Q-MCRPS SS) after application of the CSGD post-processor to the raw precipitation forecasts. The skill scores are
computed using the raw forecasts as reference. The results are presented for the three catchment groups and lead times 1, 3, and 6 days.

Figure 16 shows that, overall, improvements in precipitation forecasts are mostly associated with improvements in streamflow forecasts (points located in quadrant I; top right). In a few cases (Sys-B day 1; Sys-C and Sys-D days 3 and 6), however, improvements in precipitation forecasts are associated with negative gains in streamflow forecasts (points located in quadrant
II; top left). In some cases, improvements in precipitation are negligible, but streamflow forecasts deteriorate. These cases are mainly observed in smaller catchments, longer lead times (Day 6), and for the systems that include hydrological model structure uncertainty (Sys-C and Sys-D).

Figure 16 also reveals how the different forecasting systems interact with the precipitation post-processor to improve or deteriorate the performance of the streamflow forecasts. For example, on day 3, considering system A (Sys-A), the gain in
precipitation for a catchment pertaining to the group of smaller catchments (blue circle in the red square) does not impact the skill score of the streamflow forecasts. However, when activating the EnKF DA (e.g., Sys-B), the streamflow forecasts performance of the same catchment is improved. This improvement remains when evaluating system C and system D, although the skill score is lower (the example is indicated with red arrows in the figure). This illustrates the fact that the effect of a precipitation post-processor can be amplified if combined with the quantification of other sources of hydrological uncertainty.
A clear pattern related to catchment size is not evident, although smaller to medium-sized catchments seem to display higher skill scores for streamflow forecasts.

### 4 Discussion

From the results presented in section 3, we can answer the questions of this study: How much precipitation post-processing is needed in order to improve streamflow forecasts when dealing with a forecasting system that fully or partially quantifies
other sources of uncertainty? How does the performance of different uncertainty quantification tools compare? Finally, how does each uncertainty quantification tool contribute to improve streamflow forecast performance across different lead times and catchment sizes?

Although the precipitation post-processor undeniably improves the quality of the precipitation forecasts (Fig. 9), our results suggested that a modeling system that includes the quantification of forcing (precipitation) uncertainties only (in our case,
system A) with a precipitation post-processor is insufficient to produce reliable and accurate streamflow forecasts (Figs. 11-15). For example, in terms of bias (Fig. 11), the improvement brought by the post-processor to the worst-performing systems (systems A and C) did not outperform the other systems based on the raw precipitation forecasts. With an in-depth analysis of different configurations of forecasting systems, our study confirms previous findings that indicated that precipitation improvements do not propagate linearly and proportionally to streamflow forecasts. Interestingly, our study also shows that quantifying





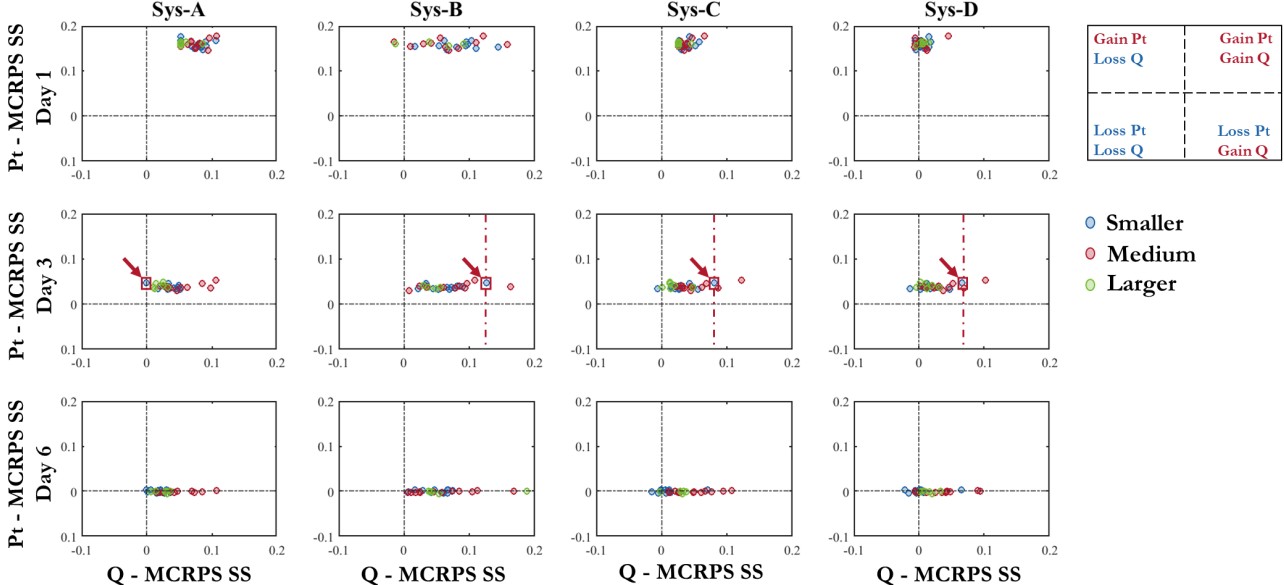

**Figure 16.** MCRPS skill score of precipitation forecasts against MCRPS skill score of streamflow forecasts after application of CSGD precipitation post-processor. The skill scores are computed using raw forecasts as reference. Results are shown for lead times of 1, 3, and 6 days and catchment group.The red box and arrows in the middle panels highlight the interaction of the precipitation post-processor with the different sources of uncertainty for the same catchment.

all sources of uncertainty (forcing, initial conditions, and model structure) does not always lead to the best results in terms of streamflow forecast performance (e.g., Fig. 15, system D), at least with the tools used in this study. Combined with precipitation post-processing, less complex systems can be good alternatives, such as those considering forcing and initial conditions uncertainty (Figs. 12 and 13, system B) and those considering forcing and multimodel uncertainty (Fig. 15, system C). This is important since current operational systems are often similar to systems B and C because they are less computationally

demanding and more prone to produce the information in a timely fashion. If the priority is to achieve a reliable and accurate system, then system B with precipitation post-processor presents a better alternative than a system like D. Additionally, the results show that the post-processor has a better interaction (provides higher gain) with the EnKF DA than with the multimodels (Fig. 16). Table 4 resumes in which circumstances a simple system with a precipitation post-processor is a best option than to quantify all sources of uncertainty.

When post-processing is not applied to precipitation forecasts, systems endowed with DA present the best performance in terms of bias and overall performance. In contrast, systems with multimodel provide better reliability. The multimodel approach was the main contribution to the ensemble spread over time (Figs. 8 and 15). Moreover, the technique's effect over time has a more significant impact than the ensemble size on forecasts performance. System B, with 2,500 members, performed similarly to system C (350 members) in terms of MCRPS as the EnKF effect fades with increasing lead times. In terms of relative bias,





**Table 4.** Partial uncertainty quantification systems (A, B and C) with precipitation post-processor against a full uncertainty quantification system (D) without post-processor in terms of BIAS, overall quality and reliability for days 1 and 6.

|  | A + CSGD | | B + CSGD | | C + CSGD | |
| --- | --- | --- | --- | --- | --- | --- |
|  | **Day 1** | **Day 6** | **Day 1** | **Day 6** | **Day 1** | **Day 6** |
| **BIAS** | ✓ | ✓ | — | ✓ | X | X |
| **Overall Quality** | X | X | ✓ | ✓ | X | — |
| **Reliability** | X | X | — | — | — | X |

(✓): simple systems with post-processor **outperform** raw-system D.

(X ): simple systems with post-processor **do not outperform** raw-system D.

(—): simple systems with post-processor have **similar performance** to raw-system D.

system A (50 members) outperformed system C, as the models did not sufficiently compensate for the error. System D (17,500 members) presented an overdispersion for the first few lead times, reducing forecast reliability.

All systems benefit from precipitation post-processing, but as expected, the effect is conditioned on lead times, forecast quality attribute, catchment size, and system configuration. The performance of the systems with multimodel was less improved by the post-processor than the systems with DA, notably for the first few lead times (Figs. 11-14). The main contribution of the

post-processor was in terms of spread for the smaller catchments and in terms of accuracy for the medium ones (Fig. 15). For the larger catchments, the effect of the post-processor in terms of spread and error is imperceptible regardless of the lead time, criteria, and system configuration and may even lose performance (Fig. 16).

## 5   Conclusion

This study aimed to decipher the interaction of a precipitation post-processor with other tools embedded in a hydrometeoro-

logical forecasting chain. We used the CSGD method as the meteorological post-processor, which yielded a full predictability distribution of the observation given the ensemble forecast. Seven lumped conceptual hydrological models were used to create a multimodel framework and estimate the model structure and parameter uncertainty. Fifty members from the EnKF and 50 members from the ECMWF ensemble precipitation forecast were used to account, respectively, for the initial conditions and forcing uncertainties. From these tools, four hydrological prediction systems were implemented to generate short- to medium-

range (1-7 days) ensemble streamflow forecasts, which vary from partial to total traditional uncertainties estimation: A) forcing, B) forcing and initial conditions, C) forcing and model structure, and D) forcing, initial conditions, and model structure. We assessed the contribution of the precipitation post-processor to the four systems for 30 catchments in the Province of Quebec, Canada, as a function of lead time and catchment size over 2011-2016. The catchments were divided up into three groups: smaller ($< 800$ km$^2$), medium (between 800 and 3,000 km$^2$), and larger ($> 3,000$ km$^2$). First, we assessed the raw precipitation

and streamflow forecasts and, in a second step, we compared them with the post-processed ones. The evaluation of the forecast



quality was carried out by implementing deterministic and probabilistic scores, which evaluate different aspects of the overall forecast quality.

The precipitation post-processor resulted in large improvements in the raw precipitation forecasts, especially in terms of relative bias and reliability. However, its effectiveness in hydrological forecasts was conditional on the forecasting system, lead time, and catchment size. Considering only meteorological uncertainty along with a post-processor may not improve hydrological forecast performance. However, quantifying all sources of uncertainty and adding a correction may worsen it. In this study, the post-processor showed a better interaction with the EnKF DA, revealing that in case all sources of uncertainties cannot be quantified, DA and a post-processor are a good alternative, especially for longer lead times.

To the best of the authors' knowledge, no previous study has explored the impact of a precipitation post-processor on a modeling chain that considers all traditional hydrometeorological sources of uncertainty. We nonetheless recognize some limitations to this study. We calibrated the CSGD parameters with the operational forecast of the ECMWF, whose model underwent improvements during the study period. Modifications to the numerical model could change the error characteristics of the forecast, affecting the efficiency of the regression model. However, despite this, the results showed that the precipitation post-processor improved the reliability of precipitation and did not affect hydrology in the same proportion. This reveals that even sophisticated post-processor techniques used in meteorology do not necessarily suit the hydrological needs.

Based on this experience, it would be interesting to consider the meteorological bias and dispersion thresholds at which hydrological predictions are affected (i.e., the meteorological error propagates significantly over the hydrology and is not mitigated by the rainfall-flow transformation process described by the hydrological models used). Future studies could also focus on determining whether calibrating the regression model parameters of the post-processor or calibrating the hydrological models with reforecasts data would lead to better results. The latter case could also serve to determine if the use of a post-processor may be avoided and how this compares to the use of a multimodel framework. Is a single calibrated model with reforecasts better than a multimodel approach? Since model C presented a good option, it would be interesting to determine if this system with a hydrological post-processor that corrects the models' bias would improve the forecasting performance without resorting to sophisticated precipitation post-processor techniques.

*Code and data availability.* All tools used in this study are open to the public. The software used to build the forecasting systems is available in a GibHut repository (https://github.com/AntoineThiboult/HOOPLA). ECMWF precipitation data used in this study can be obtained freely from the TIGGE data portal (https://www.ecmwf.int/en/ research/projects/tigge). The observed datasets were provided by The Direction d'Expertise Hydrique de Québec and can be obtained on request for research purposes.





## Appendix A: Verification Metrics

### A1  Relative Bias (BIAS)

$$BIAS = \frac{\sum_{k=1}^{N} Fct_{avg}(k)}{\sum_{k=1}^{N} Obs(k)} \tag{A1}$$

where $(Fct_{avg}(k), Obs(k))$ is the $k^{th}$ of $N$ pairs of deterministic forecasts and observations.

### A2  Continuous Ranked Probability Score

$$CRPS(K) = \int_{-\infty}^{\infty} \left[ F_k^{'}(x) - H(x \geq x_{obs}) \right]^2 dx \tag{A2}$$

where $F_k(x)$ is the cumulative distribution function of the $k^{th}$ realization, $x$ is the predicted variable, and $x_{obs}$ is the corresponding observed value. $H$ is the Heaviside function, which equals 0 for predicted values smaller than the observed value, 1 otherwise.

### A3  Spread-skill plot (SSP)

$$RMSE = \left[ \frac{1}{N} \sum_{k=1}^{N} (Fct(k) - Obs(k))^2 \right]^{1/2} \tag{A3}$$

$$SPREAD = \sqrt{\frac{1}{N} \sum_{k=1}^{N} \frac{1}{M-1} \sum_{j=1}^{M} (\overline{Fct}_k - Fct_{j,k})^2} \tag{A4}$$

where M is the number of members. The RMSE retains the units of the forecast variable, and the lower its value, the better.

### A4  Interquantile Range (IQR)

$$IQR = \frac{1}{N} \sum_{k=1}^{N} Fct^{95}(k) - Fct^{05}(k) \tag{A5}$$

where $(Fct^{95}(k), Fct^{05}(k))$ is the $k^{th}$ of $N$ pairs of quantiles of the forecasts.

### A5  Volumetric efficiency (VE)

$$VE = 1 - \frac{\sum_{k=1}^{N} |Q_{sim}(k) - Q_{obs}(k)|}{\sum_{k=1}^{N} Q_{obs}(k)} \tag{A6}$$

where $(Q_{sim}(k)$ and $Q_{obs}(k))$ is the $k^{th}$ of $N$ pairs of simulated and observed streamflows.



## A6  Modified Kling-Gupta efficiency

$$KGE = \sqrt{(r-1)^2 + (\beta-1)^2 + (\gamma-1)^2} \tag{A7}$$

$$\beta = \frac{\mu_{sim}}{\mu_{obs}} \tag{A8}$$

$$\gamma = \frac{CV_{sim}}{CV_{sim}} = \frac{\sigma_{sim}/\mu_{sim}}{\sigma_{obs}/\mu_{obs}} \tag{A9}$$

where $r$ is the correlation coefficient reflecting the linear relationship between simulation (the ensemble mean in forecasting mode) and observation. $\mu_{sim}$ ($\mu obs$) and $\sigma_{sim}$ are the mean and the standard deviations of the simulated (observed) time series. CV is the coefficient of variation.

*Author contributions.*  All authors contributed to designing the experiment. EV conducted the numerical experiments, led the results analysis and the production of the figures. FA and MHR supervised the study and contributed to the interpretation of results. FA was responsible for funding acquisition. EV wrote the paper, and FA and MHR provided input on the paper for revision before submission.

*Competing interests.*  The authors declare that they have no conflict of interest.

*Acknowledgements.*  Funding for this work was provided to the first and second authors by FloodNet, an NSERC Canadian Strategic Network
(Grant number: NETGP 451456-13), and by NSERC Discovery Grant RGPIN-2020-04286. The authors thank the Direction d'Expertise Hydrique du Québec for providing hydrometeorological data and ECMWF for maintaining the TIGGE data portal and providing free access to archived meteorological ensemble forecasts. Special acknowledgments go to Dr. Michael Scheuerer for sharing the R codes of the CSGD processor and offering many insights on CSGD.



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
