# Peer review of "Choosing between post-processing precipitation forecasts or chaining several uncertainty quantification tools in hydrological forecasting systems"

_Hydrology and Earth System Sciences, 2021_

## Referee Comment (RC1)

The authors analyzed the effect of precipitation post-processor (CSGD) for four different four hydro-meteorological forecasting systems. Those forecasting systems were differed by the degree of uncertainty considered: system A) forcing, system B) forcing and initial conditions, system C) forcing and model structure, and system D) forcing, initial conditions, and model structure. The results showed that precipitation post-processor worked better for less complex hydrological systems (system A/B/C). Quantifying all sources of uncertainty (forcing, initial conditions, and model structure) did not always lead to the best results in streamflow forecasting. The authors also compared the contribution of the post-processor for different catchment sizes.

The subject clearly fits into the scope of the journal and provides a useful guidance for the choice of forecasting systems. The authors sufficiently draw upon the existing body of literature and the research is interesting. However, I have some concerns with the research method. For these reasons, I recommend a decision of Minor Revision for this manuscript. Please find my comments in the following. I hope you find them useful.

1. The research is a comparison between different forecasting systems. Quantitative results are needed to justify the conclusions. However, the authors tend to use sentences like 'Line 467-468: We also observe that post-processing precipitation forecasts have a **much higher** impact on the quality of precipitation forecasts (Fig. 9) than on the quality of streamflow forecasts, as evaluated by the BIAS score', or 'Line 491-491: It is interesting to note that, for systems B, C, and D, streamflow forecasts based on raw precipitation forecasts are always **much better** than streamflow forecasts based on post-processed precipitation forecasts in system'. They don't give the readers an objective description for the research. The qualitative results cannot help the scientific choice. Please provide more numerical results, especially in conclusion.

2. Concerns about the multimodel approaches. From Figure 8, the multimodel approach seems to bring additional uncertainty to streamflow forecasts, as system C always has the worst results in term of BIAS and IQR. From Figure 5, the models are with an average KGEm of 0.64 in validation without EnKF. This is a quite low value. When the hydrological uncertainty is dominant, it is difficult to analyze the effect from precipitation post-processor. So, the boxplot for system C with or without post-processor for lead time 1 day in Figure 11 is similar. It is not indicated that the precipitation post-processor brings in no improvements. The improvements probably are too minor to offset the hydrological errors.

3. The author's language usage was difficult to read at times. Too many adverbial clauses and attributive clauses make the sentence too long to understand.

4. In Section 2.1.1, the authors reduced the ECMWF database to 0.1° and then spatially averaged forecasts to the catchment scale. I am confused by the resolution reduction as the catchment areal forecasts were used. It is more simple to use the archived 0.25° to calculate the catchment average. The resolution reduction might bring in additional

uncertainty to precipitation forecasts.

5. Line 170-171: The authors mentioned they used an adapted CSGD to post-process ECMWF precipitation. Whether the only difference is that the original CSGD used neighboring grid points while they used all grids in a catchment?

6. Line 221-222: the same….as….

7. Line 506-509: NWP products often fail to capture precipitation forecasts in small domain, yet behave better in large catchments. Lumped hydrological models are more likely to better model streamflow at small basins, where the hydrological process is simpler and easier to be simulated by those simple lumped models.

8. Since the authors elaborately analyze the performance for different catchment size in Section 3.4, they should provide detailed validation results of the hydrological models in different catchments in Section 3.1. It would help the readers to understand the forecast skill over catchment sizes.

---

## Author Comment (AC1)

The authors would like to thank the Anonymous Referee #1 for the detailed review of our paper and the many constructive comments. We believe that they will enhance the document significantly. In the following, we provide our answers. The reviewer comments are printed in black and our replies, in blue.

**RC1: Anonymous Referee #1**

1. **RC1:** *The research is a comparison between different forecasting systems. Quantitative results are needed to justify the conclusions. However, the authors tend to use sentences like 'Line 467-468: We also observe that post-processing precipitation forecasts have a **much higher** impact on the quality of precipitation forecasts (Fig. 9) than on the quality of streamflow forecasts, as evaluated by the BIAS score', or 'Line 491-491: It is interesting to note that, for systems B, C, and D, streamflow forecasts based on raw precipitation forecasts are always **much better** than streamflow forecasts based on post-processed precipitation forecasts in system'. They don't give the readers an objective description for the research. The qualitative results cannot help the scientific choice. Please provide more numerical results, especially in conclusion.*

   **Authors Replay (AR):** We will revise the description of the results to provide a more in depth numerical analysis.

2. **RC1:** *Concerns about the multimodel approaches. From Figure 8, the multimodel approach seems to bring additional uncertainty to streamflow forecasts, as system C always has the worst results in term of BIAS and IQR. From Figure 5, the models are with an average KGEm of 0.64 in validation without EnKF. This is a quite low value. When the hydrological uncertainty is dominant, it is difficult to analyze the effect from precipitation post-processor. So, the boxplot for system C with or without post-processor for lead time 1 day in Figure 11 is similar. It is not indicated that the precipitation post-processor brings in no improvements. The improvements probably are too minor to offset the hydrological errors.*

   **AR:** This is a very interesting point, and we thank the reviewer for drawing attention to it.

   The 0.64 is the mean of all models in all catchments considered separately, not as a multimodel. That is, the performance of each of the models was determined individually and then averaged. In the case of the multimodel, its average yield is 0.73 (see figure attached to question 12 of reviewer #2). The simulations of each model are considered as an ensemble and then their performance is determined.
   It is also good to remember that extreme values affect the mean. As shown in Figure 5, models 1, 4, 6, and 7 experience difficulties in simulating some basins, which is to be expected since no single model excels in all situations. In the case of systems using only one model, the one used is the median during calibration. As we responded to reviewer 2, it is almost impossible to predict which model will be the best predictor on any given day and basin, and that is when a multimodel has value.

In the revised version, we will add the performance of the multimodel in Figure 5 to avoid confusion, and we will also soften our wording to recognize the value of the post-treatment. This point raised by the reviewer is worth mentioning, and we will include a comment on it.

3. **RC1:** *The author's language usage was difficult to read at times. Too many adverbial clauses and attributive clauses make the sentence too long to understand.*

   **AR:** Fixing this will be a priority when producing a revised version.

4. **RC1:** *In Section 2.1.1, the authors reduced the ECMWF database to 0.1° and then spatially averaged forecasts to the catchment scale. I am confused by the resolution reduction as the catchment areal forecasts were used. It is more simple to use the archived 0.25° to calculate the catchment average. The resolution reduction might bring in additional uncertainty to precipitation forecasts.*

   **AR:** We fully agree with the reviewer that resolution reduction might bring additional uncertainty to precipitation forecasts. However, the original resolution of the ECMWF is too coarse for this application, especially for the smaller group of catchments (11 in total). The surface of this group is less than 800 km2, and in many cases, only one meteorological forecast grid point falls within catchment boundaries and in others none. Therefore, downscaling ensures that several points fall within the catchment boundaries.

   In addition, Thiboult et al. 2016 suggested that when we reduce the spatial resolution, we consider the contribution of the points close to the catchment boundaries, which allows us to have a better description of the meteorological conditions of the catchments. This is also corroborated by Scheuerer and Hamill 2015a who demonstrated that it is beneficial to add forecasts from grid points within a certain neighborhood of the location of interest as potential predictors to account for position uncertainty.

   We will clarify the purpose behind the downscaling in the revised version.

5. **RC1:** *Line 170-171: The authors mentioned they used an adapted CSGD to post-process ECMWF precipitation. Whether the only difference is that the original CSGD used neighboring grid points while they used all grids in a catchment?*

   **AR:** The reviewer understood this issue correctly. The original CSGD uses neighborhood information from grid-based forecasts to compute the ensemble statistics, thus accounting for displacement errors. In our case, we computed the ensemble statistics directly from the mean areal precipitation over catchments. As mentioned in the previous answer, the downscaling implicitly allows considering the contribution from neighboring grids.

6. **RC1:** *Line 221-222: the same….as….*

   **AR:** We will correct in the revised version

7. **RC1:** *Line 506-509: NWP products often fail to capture precipitation forecasts in small domain, yet behave better in large catchments. Lumped hydrological models are more likely to better model streamflow at small basins, where the hydrological process is simpler and easier to be simulated by those simple lumped models.*

   **AR:** Thank you for pointing this out. As suggested by Reviewer # 2, we will shift the emphasis to the properties of the catchments rather than the modeling approach (see our answer to question 13 of Reviewer #2 for details).

8. **RC1:** *Since the authors elaborately analyze the performance for different catchment size in Section 3.4, they should provide detailed validation results of the hydrological models in different catchments in Section 3.1. It would help the readers to understand the forecast skill over catchment sizes.*

   **AR:** Thank you very much for this suggestion. We propose to include the figure below and move Section 3.1 to the supplementary material as recommended by Reviewer #2 to decrease the length of the paper.

[Figure]

**References**

Thiboult, A., Anctil, F., and Boucher, M.-A.: Accounting for three sources of uncertainty in ensemble hydrological forecasting, Hydrol. Earth Syst. Sci., 20, 1809–1825, https://doi.org/10.5194/hess-20-1809-2016, 2016.

Scheuerer, M. and Hamill, T. M.: Statistical postprocessing of ensemble precipitation forecasts by fitting censored, shifted gamma distributions, Monthly Weather Review, 143, 4578–4596, https://doi.org/https://doi.org/10.1175/MWR-D-15-0061.1, 2015.

---

## Author Comment (AC2)

The authors would like to thank the Anonymous Referee #2 for their detailed review of our paper and the many constructive and interesting comments. We believe they will enhance the document significantly. In the following, we provide our answers. The reviewer comments are printed in black, and our replies, in blue.

**RC2: Anonymous Referee #2**

**Major comments:**

1. **RC2:** *Figure 8, and the commentary associated with it, is problematic. The authors are comparing reliability of their hydrological uncertainty quantification methods when they know that their forcings are unreliable (as deftly shown in fig 10). This makes assessments of reliability meaningless: the streamflow forecasts in fact \*should\* be unreliable, as the forcings are unreliable. So, for example, in Fig 8 System B looks bad next to System D (for reliability), but Fig 13 shows that when forcings are reliable, System B performs similarly to System D. So Fig 8 gives misleading information - indeed, one could argue it shows System B to be superior (despite the appearance to the contrary). There are a few ways around this - e.g. the authors could generate hydrological 'forecasts' that are forced with observations to produce Fig 8 - but the simplest way is to remove this figure and the accompanying discussion, and focus more on Figs 11-14.*

    **Authors Replay (AR):** This is a relevant remark, and we understand the reviewer's point. Unfortunately, current operational systems are often like systems A, B, and C. We wanted to overview how each tool contributes to the forecast performance over time and how a post-processor can improve it. This highlights three important aspects: 1) the need to correct the inputs before being used, 2) how reliability increases as we quantify other sources of uncertainty, and 3) the fact that systems like Sys-A could lead to suboptimal decisions. Furthermore, as suggested, we used the observations to explore a new figure similar to figure 8, and the conclusions are the same (Syst-A is the least reliable). Accordingly, we propose to keep figure 8 and highlight the aspects pointed out by the reviewer.

2. **RC2:** *The paper is admirably comprehensive, but perhaps as a result somewhat long. Longer papers are less likely to be read, and I really feel like this paper deserves to be read widely! So I have a few suggestions for shortening it:*

    **AR:** Thank you for your comments that greatly contribute to improving our paper.

    *i) Figure 7 replicates information in Figure 9. I suggest omitting Figure 7 and restructuring the paper to first discuss rainfall forecasts (both raw and post-processed) and then streamflow forecasts.*

    **AR:** Following the reviewer's suggestion, we propose the following structure for the results section (Section 3)

3. Results

    3.1 Ensemble precipitation forecasts (Figures 9 and 10)

        3.1.1 Performance of raw precipitation forecasts

        3.1.2 Performance of corrected precipitation forecasts

    3.2 Ensemble streamflow forecasts (Figures 11-14)

        3.2.1 Performance of raw streamflow forecasts

        3.2.2 Interaction of the precipitation post-processor with the hydrological forecasting systems

    3.3 Effect of catchment size (new Figures based on the MAErd and the MCRPS)

        3.3.1 Gain in streamflow forecasts from CSGD correction

*ii) Fig 8 should be omitted, as noted above*

**AR:** See the answer to question 1.

*iii) Figs 11-14 could be combined into a single figure. At the very least, I suggest Figs 13 & 14 should be combined - as the authors note, sharpness without reliability is meaningless.*

**AR:** We thank the reviewer for the suggestion. We will use the following figures in the revised version:

[Figure]

**Figure 7.** Relative bias (BIAS) and MCRPS of the ensemble streamflow forecasts of the four hydrological prediction systems and lead times 1, 3, and 6 days when considering raw (blue) and post-processed (red) precipitation forecasts. Boxplots represent the score variability over the 30 catchments.

[Figure]

**Figure 8.** MAE of the Reliability Diagram (MAErd) and Interquartile Range (IQR) of the ensemble streamflow forecasts of the four hydrological prediction systems and lead times 1, 3, and 6 days when considering raw (blue) and post-processed (red) precipitation forecasts. Boxplots represent the score variability over the 30 catchments.

*iv) [Optional] The discussion of the hydrological model performance (Section 3.1) and the accompanying figure, while interesting, could be moved to supplementary materials.*

**AR:** Thank you for this suggestion. We will move Section 3.1 to supplementary material and add another figure with detailed validation results of the hydrological models for the different catchment groups as suggested by reviewer #1.

3. **RC2:** *Figure 15 shows results that appear to be somewhat inconsistent with Figs 12 & 13. For example, in Figure 15 Sys-C looks best (most reliable) across all catchments compared to other systems - much better than Sys-B or Sys-D. This is very much not the case in Fig 13, where Sys-B & Sys-D are more reliable when paired with the CSGD forcing. Further, the RMSE values for the CSGD forcings are often almost identical to the raw forcings, which is not consistent with the CRPS scores presented in Fig 12. I think spread-skill plots are a pretty rough way of looking at reliability (they reduce the 'spread' of the ensemble to root of the average variance, rather than considering the full distribution, e.g. like Probability Integral Transforms/Rank Histograms do), and I suspect the reliability diagram analyses presented by the authors in Fig 13 give a better picture. It may be better to simply use MAE_rd & CRPS in Figure 15 for consistency with Figs 12 and 13. If the authors keep the spread-skill plots in Fig 15 (they may have different views to me on the efficacy of the different reliability assessments) - I think the apparent discrepancies between Fig 15 and Figs 12/13 should be explained.*

**AR:** This is a relevant remark. We will follow the reviewer's suggestions and use the MAErd for figure 15 in the revised version. In this way, we will avoid the multiplication of indicators that sometimes react differently. We will then be consistent with the other figures, and it will help reduce the paper's length. However, it is good to note that in Figure 15, we are evaluating reliability by watershed group (lines present the median) as opposed to Figures 11-14, where the boxplots include all watersheds.

On the other hand, the RMSE is larger in the raw than in the corrected forecasts in the medium catchment group. Moreover, the RMSE value is larger in system C than in the others. This is congruent with Figure 12. There should indeed be some congruence between the MCRPS and RMSE, but both evaluate different attributes of the forecast quality.

**Specific (minor) comments:**

1. **RC2:** *L26-27 'The inherent uncertainty of hydrological forecasts stems from three main sources' there is a crucial fourth source the authors have omitted: uncertainty in observations.*

   **AR:** Definitively, the reviewer is right. We will add this source of uncertainty in the revised version.

2. **RC2:** *L28 "Two main philosophies are generally adopted" I understand why the authors are making this distinction, but I guess I would argue the emergence of a third class of systems, namely hybrid statistical-dynamical forecasting systems - which combine physical models (e.g. ensemble NWP) with statistical processing methods (e.g. NWP calibration) - is worth mentioning. Statistical post-processors (hydrological uncertainty processors, ensemble dressing) are very effective at correcting statistical aspects of forecasts (bias, uncertainty quantification), but usually produce discrete probability distributions in time (e.g. one for each lead time) and space (e.g. one for each gauge) - they are not connected in time and space. Users of hydrological forecasts are often interested in hydrographs, which require coherent information in time (across lead times) and (often) in space (e.g. upstream and downstream gauges). Coherent spatial and temporal information allows users to, for example (i) check flood peaks as well as total volume forecasts for, say, the next week; (ii) when flood peaks are expected to pass by a sequence of gauges. Ensemble systems can present all this information, but when they are based on purely physical/conceptual models they often do not get statistical aspects of forecasts (bias, uncertainty quantification) correct. Hybrid statistical-dynamical systems are in their infancy, but they promise the best of both purely statistical or purely dynamical ensmeble systems.*

   **AR:** Thank you very much for the suggestion. We will include this valuable information in the new version.

3. **RC2:** *L71-73 "Hydrological post-processing ... cannot compensate for weather forecasts that are highly biased" Yes, it can - though it depends on the method employed. Hydrological uncertainty*

*processors (e.g. Biondi and Todini 2018) lump all sources of uncertainty - including biases in rainfall forecasts - into the model, and they are effective at correcting these issues. They are not so good if you need each ensemble member to represent a temporally coherent hydrograph, however.*

**AR:** The reviewer is right. Some hydrological post-processing techniques have been developed to directly address the system's total uncertainty (Demirel et al.2013, Kavetski et al.2006) but this is a subject still in its infancy and, to our knowledge, have not been implemented in an operational context. In many cases, the results are synthetic series that represent a probability and not a real forecast. As the reviewer points out here and in question 2, the temporal representation of each member may be important if the ultimate goal is to feed another model (e.g., hydraulic). This lack of consistency is likely to generate a different response from the catchment (faster/slower flood, event duration, etc.) and is not desirable. We propose to add this:

"Hydrological post-processing … cannot compensate for weather forecasts that are highly biased (Abaza et al., 2017; Roulin and Vannitsem, 2015) unless it addresses the many sources of uncertainty in an integrated manner (Biondi and Todini 2018, Demirel et al.2013, Kavetski et al.2006)."

4. **RC2:** *L84 "Considering several sources of uncertainty in ensemble hydrometeorological forecasting inevitably implies an increase in the number of simulations" This is not 'inevitable' - e.g. Bennett et al. (2016), who add hydrological uncertainty to rainfall forecast uncertainty without increasing the number of ensemble members. Suggest this be changed to "Considering several sources of uncertainty in ensemble hydrometeorological forecasting often implies an increase in the number of simulations".*

**AR:** Thank you for the remark. We will modify this in the revised version.

5. **RC2:** *L87 "However, there is also additional uncertainty associated with the assumptions made in creating a larger ensemble." Please be explicit about why increasing ensemble size is a problem.*

**AR:** In this case, we refer to ontological uncertainty (uncertainty associated with different belief systems). As highlighted by (Beven, 2016), different beliefs about the appropriate assumptions could lead to very different uncertainty estimates. Therefore, we propose to modify the paragraph as follows:

"However, there is also additional uncertainty associated with the assumptions made in creating a larger ensemble. A plethora of methods exists to quantify each source of uncertainty, which implies ontological uncertainties (Beven, 2016). If the methodologies implemented to simulate the sources of forecast error and quantify various uncertainty sources are inappropriate, even a large ensemble size could be under or over dispersive and thus not represent the total predictive uncertainty accurately (Boelee et al., 2019; Buizza et al., 2005)."

6. **_RC2:_** _L137 "modeling uncertainty" should this be "hydrological modeling uncertainty"?_

   **AR:** We will correct this in the revised version.

7. **_RC2:_** _L211-213 "As such, it relaxes the constraint of parametric assumptions (difficult for variables like precipitation) and high dimensional cases (Gneiting, 2014). This technique uses the raw ensemble forecasts trajectories to identify the dependence template. Therefore, the post-processed ensemble should have the same number of members as the raw ensemble." I didn't find this explanation very clear or helpful. ECC enforces rank correlations (both temporal and spatial) from the raw forecasts in the calibrated forecasts, much like the Schaake shuffle. I would simply say something like this._

   **AR:** Thank you very much for this suggestion. We propose the following:

   "This technique reconstructs the spatio-temporal correlation _by reordering samples from the raw predictive marginal distributions_ to identify the dependence template. The ECC reasoning lies in the fact that the physical model can adequately represent the covariability between the different dimensions (i.e., space, time, variables). Furthermore, as the template is the raw ensemble, the ECC should have the same number of members as the raw ensemble."

8. **_RC2:_** _L219 "provide model state" The authors use a range of models (nwp, statistical, hydrological), and it would be helpful to clearly identify (here and elsewhere) which ones they are talking about in each case, rather than referring to generic 'models'. Here I think the authors are discussing hydrological model states._

   **AR:** The reviewer is right, in L219 we are discussing hydrological model states. We will identify which model we are referring to throughout the paper in the revised version.

9. **_RC2:_** _L315 "Root Mean Square Error" is this calculated on the ensemble mean? (And if not, how is it calculated?)_

   **AR:** Yes, it is calculated on the ensemble mean. However, as suggested by the reviewer, we will make Figure 15 based on the MAErd to shorten the paper and maintain consistency with the other figures. All information related to the spread-skill plot will be removed in the next version.

10. **_RC2:_** _L333-334 "Accordingly, we decided to use the reliability diagram to evaluate precipitation for thresholds of different exceedance probabilities (EP), namely 0.05, 0.5, 0.75, and 0.95". Could the authors please confirm 1) that to construct the reliability diagrams, they have converted probabilistic forecasts to binary predictions for exceeding thresholds based on these quantiles and 2) these quantiles are taken from observations and 3) that they have calculated MAE_rd across all these thresholds? These things weren't entirely clear to me from the description. L338 IQR (interquartile range) usually denotes the range between two quartiles (25%ile and 75%ile). So_

*'IQR' is not the correct term to use for a different interval. Suggest using the term 'average width of prediction interval' (AWPI).*

**AR:**

1) Yes. The reviewer is correct.

2) Yes, those quantiles were taken from the observations.

3) No, we just used those quantiles for Figure 10. To compute the MAErd, we calculate nine confidence intervals with nominal confidence level of 10–90%, with an increment of 10% for each emitted forecast. Then, it was established whether or not each confidence interval covered the observation for each forecast and each confidence interval. This was repeated for all the forecast–observation pairs. In a well-calibrated distribution, the observation inside each confidence interval and its corresponding nominal confidence level should be close, taking the form of a linear relationship 1:1 (as the reliability diagram). Finally, the difference between these values is equivalent to the MAE of the reliability diagram. In this way, we treat the forecasts as continuous variables.

In L338 it was a typo error. The correct word is Interquantile, as shown in Figure 4. We used the 90% Interquantile range, the difference between the 95th quantile and the 5th quantile (Gneiting et al. 2007) that have been used widely as a metric to evaluate the sharpness in the hydrometeorological forecasting community (Cassagnole et al. 2021, Bourgin et al. 2014, Crochemore et al., 2016).

We will clarify all these points in the revised version.

11. **_RC2:_** *L362 Figure 4 - this is a really nice figure!*

**AR:** Thank you! 😊

12. **_RC2:_** *L386 Figure 5 - would be helpful to have abbreviated names of the models along the xaxis rather than M1, M2, etc, to avoid having to track back to Table 2 to interpret this figure. These only need to appear on the bottom panel.*

**AR:** We will use this figure in the revised version (caption remains the same):

[Figure]

13. **RC2:** *L507 "This is particularly observed in the streamflow forecasts and may be related to the use of lumped hydrological models, which tend to smooth errors over larger modeling areas" This is kind of saying the same thing, but I'd shift the emphasis to the properties of the catchments themselves, rather than lumped hydrological models. Something like "For a given time step, smaller catchments tend to show more variance and are thus more difficult to forecast than larger catchments." (I.e., it's the larger catchments themselves that smooth, e.g., spatial and temporal variability in rainfall; the models just try to replicat this.)*

**AR:** We thank the reviewer for the suggestion. Reviewer #1 also pointed out this. We will modify it accordingly in the revised version. We propose to add on L507:

"This is particularly observed in the streamflow forecasts and may be related to the fact that larger catchments tend to experience lower streamflow variability (Sivapalan, 2003) and are thus easier for the model to reproduce. Moreover, weather can differ substantially over a couple of kilometers, and numerical model resolution is too coarse to capture these variations in smaller catchments. Thus, for example, it would miss extreme events even if the amount of precipitation is well predicted but in the wrong place. However, in larger catchments, a buffer effect can be generated on the total streamflow."

14. **RC2:** *L515-516 "These findings confirm that large improvements in precipitation forecasts do not necessarily lead to improvements in streamflow forecasts." I don't wholly agree with this. First, Sys-B \*does\* show marked improvement in reliability when it is forced with reliable inputs. So*

*there is improvement there. The 2 best peforming methods, to my eye, are Sys-B and Sys-D when forced with Q_CSGD. Sys- B is marginally superior: at Day 6, Sys-B with Q_CSGD is sharper than Sys-D, despite being similarly reliable. (When you consider the additional complexity of Sys-D, Sys-B to my mind is clearly preferable for operational forecasting.) Further, the measures of reliability the authors have chosen may not be all that sensitive. I personally prefer the use of probability integral transforms (PIT), or their summary statistics (e.g. Renard et al.'s (2010) 'alpha' statistic - very similar to the authors' MAE_rd). Because PIT treats forecasts as continuous variables, there's no need to simplify forecasts to binary predictions of thresholds. I'm not suggesting any changes here - the analysis the authors have presented is easily thorough enough, but I think they are understating the value of the CSGD in the system, and should probably soften their wording here.*

*__AR:__ In L515-516 we refer particularly to smaller catchments, not in a general sense. Improvements brought by the post-processor were more significant in terms of RMSE in precipitation than in terms of Spread. The opposite was true for streamflow. We recognize the value of CSGD. That is why we generally said: "not necessarily." In fact, in Table 4, we highlight the systems that, together with the post-processor, could be equal or better than system D. We conclude that the effectiveness is conditioned to several factors, and one of them is the catchment's size. We will be sure to emphasize the value of the CSGD in the corrected version.*

15. __RC2:__ *L537 "In system D, the application of precipitation post-processing does not lead to reliable streamflow forecasts: the RMSE and spread curves in red in Fig. 15 are not aligned, while the blue curves (for raw forecasts) are for lead times greater than 3 days" The reason seems clear: Sys-D ensembles are wider than they need to be at Day 6, as shown cf Sys-B in Fig 14. So while they are reliable, they are not as sharp as they could be. In short, Sys-D overestimates uncertainty at long lead times.*

    *__AR:__ The reviewer is correct. But as mentioned above, all spread-skill plot related figures and analyses will be removed in the revised version.*

16. __RC2:__ *L574 Fig 16 - this is a fascinating figure, probably worth a paper on its own. Great stuff!*

    *__AR:__* Thank you! 😊

17. __RC2:__ *L578 "(Fig 15 system C)" As already noted, I have misgivings about Fig 15. In other figures (Figs 11-14) Sys-C is clearly outperformed by both Sys-B and Sys-D. And the reason is not really surprising: hydrological models, while somewhat diverse in structure (in some respects they can be quite similar, too), often perform very similarly once calibrated (as the authors themselves show in Section 3.1). So the uncertainty explained by a range of calibrated models is likely to be*

*quite small, explaining why Sys-B can outperform Sys-C even though it uses a single hydrological model.*

**AR:** *We do not wholly agree with this. Figure 5 shows that the models react differently from one location to another (boxplot variability). Models 2, 3, and 5 show a similar range, but this is not the case for models 1, 4, 6, and 7. Numerous papers confirm that parameter uncertainty is accounted for when a multi-model approach is used. The figure attached to question 12 makes this clear. We can see that a multi-model approach is more beneficial than using models 1, 4, 6, and 7 individually. Some models simulate some events better than others, so their performance varies during a hydrological year. We know that it is a nightmare and almost impossible to predict which model will be the best predictor on any given day and that is when the multi-model approach has its value.*

18. **RC2:** *L581 "If the priority is to achieve a reliable and accurate system, then system B with precipitation post-processor presents a better alternative than a system like D." Yes, I fully agree.*

**AR:** Thank you! 😊

19. **RC2:** *L586 "When post-processing is not applied to precipitation forecasts... systems with multimodel provide better reliability" I agree that this is what Figs 8 and 15 show, but as discussed: Fig 8 shouldn't be expected to produce reliable forecasts, and Fig 15 appears to be at least partly inconsistent with Figs 12 & 13. These conclusions can only be drawn with certainty if Fig 8 was constructed from streamflow forecasts generated with 'perfect' (observed) rainfall forcings to isolate the effects of the hydrological uncertainty generation methods. I don't think this paper needs this additional analyses - it has plenty of material already - so I suggest omitting this paragraph.*

**AR:** *As mentioned in question 1, even using perfect forecasts, the conclusions remain the same. The hydrological ensemble variation is substantially larger and contributes more actively over time to reliability than the meteorology or initial conditions ensembles.*

20. **RC2:** *L589-590 "System B, with 2,500 members, performed similarly to system C (350 members) in terms of MCRPS as the EnKF effect fades with increasing lead times." I hadn't realised the difference in ensemble size was so stark. It's likely that the discrepancy in ensemble sizes would have impacted on all their calculations (though I accept the authors' contention that the effect was small) - however for future work I suggest subsampling from the large ensembles to control for ensemble size, or using unbiased estimators that control for ensemble size (e.g. Ferro et al. (2008)) so any potential artifacts of ensemble size are totally removed.*

**AR:** As mentioned in L202-203, we are aware that the ensemble size may generate bias implying that the apparent superiority of a system is due solely to its larger ensemble size and not to differences in modeling. Most studies conclude that increasing the ensemble size improves performance. However, that was not always the case in our study. Moreover, as pointed out by Buizza and Palmer 1998, performance improvement brought by increasing the ensemble size is highly dependent on the metric used to evaluate performance.

We explored the correction factor proposed by Ferro et al. 2008 but seeing that the difference was not substantial (the difference was in the 3rd decimal), we decided to use the standard MCRPS to emphasize that the technique used is more significant than the ensemble size, as expressed by Sharma et al. 2019.

21. **RC2:** *L630 "Since model C presented a good option, it would be interesting to determine if this system with a hydrological post-processor that corrects the models' bias would improve the forecasting performance without resorting to sophisticated precipitation post-processor techniques." To reiterate, the conclusion that System C is a good option is really based only on Fig 15, about which I have misgivings. If these misgivings are borne out, I would suggest that System C would not necessarily be considered a 'good option'.*

    **AR:** *It is true that in most of the forecast attributes evaluated, system C presents a lower performance than systems B and D. However, system C presents a better MAErd than system B, especially on the farthest horizons (Figure 13). Perhaps "good option" is not appropriate to use, and it should be specified that this is in terms of reliability. We propose:*

    *"Since model C presented a better reliability than systems A and B, it would be interesting to determine if this system, with a hydrological post-processor that corrects the models' bias, would improve the forecasting performance without resorting to sophisticated precipitation post-processor techniques."*

**Typos etc.**

*L38 "parameters uncertainty" should be "parameter uncertainty"*
Will be corrected in the revised version

*L101 "directly use NWPs outputs" should be "directly use NWP outputs"*

Will be corrected as well in the revised version

**References**

Sivapalan, M. (2003), Process complexity at hillslope scale, process simplicity at the watershed scale: Is there a connection?, Hydrol. Processes, 17, 1037–1041, doi:10.1002/hyp.5109.

Kavetski D, Kuczera G, Franks SW. Bayesian analysis of input uncertainty in hydrological modeling: 2. Application. Water Resour Res 2006, 42:W03408. https://doi.org/10.1029/2005WR004376

Demirel MC, Booij MJ, Hoekstra AY. Effect of dif- ferent uncertainty sources on the skill of 10 day ensemble low flow forecasts for two hydrological models. Water Resour Res 2013, 49:4035–4053.

Gneiting, T., Balabdaoui, F., and Raftery, A.: Probabilistic forecasts, calibration and sharpness, J. Roy. Stat. Soc. B, 69, 243–268, https://doi.org/10.1111/j.1467-9868.2007.00587.x, 2007.

Crochemore, L., Ramos, M.-H., and Pappenberger, F.: Bias correcting precipitation forecasts to improve the skill of seasonal streamflow forecasts, Hydrology and Earth System Sciences, 20, 3601–3618, https://doi.org/10.5194/hess-20-3601-2016, 2016.

Zalachori, I., Ramos, M.-H., Garçon, R., Mathevet, T., and Gailhard, J.: Statistical processing of forecasts for hydrological ensemble prediction: a comparative study of different bias correction strategies, Advances in Science and Research, 8, 135–141, https://doi.org/10.5194/asr-8-135-2012, 2012.

Bourgin, F., Ramos, M. H., Thirel, G., and Andréassian, V.: Investigating the interactions between data assimilation and post-processing in hydrological ensemble forecasting, Journal of Hydrology, 519, 2775–2784, https://doi.org/10.1016/j.jhydrol.2014.07.054, 2014.

Cassagnole, M., Ramos, M.-H., Zalachori, I., Thirel, G., Garçon, R., Gailhard, J., and Ouillon, T.: Impact of the quality of hydrological forecasts on the management and revenue of hydroelectric reservoirs – a conceptual approach, Hydrology and Earth System Sciences, 25, 1033–1052, https://doi.org/10.5194/hess-25-1033-2021, 2021.

---

## Author Response (AR1)

The authors would like to thank the two anonymous Referees for the detailed reviews of our paper and the many constructive and interesting comments. We believe they enhanced the document significantly, and we hope that the revised version meets the requirements to be published at Hydrology and Earth System Sciences. In the following, we provide a point-by-point reply to each of the comments and an explanation of how we included them in the text. The reviewer comments are printed in black, and our replies in blue.

**RC1: Anonymous Referee #1**

1. **RC1:** *The research is a comparison between different forecasting systems. Quantitative results are needed to justify the conclusions. However, the authors tend to use sentences like 'Line 467-468: We also observe that post-processing precipitation forecasts have a **much higher** impact on the quality of precipitation forecasts (Fig. 9) than on the quality of streamflow forecasts, as evaluated by the BIAS score', or 'Line 491-491: It is interesting to note that, for systems B, C, and D, streamflow forecasts based on raw precipitation forecasts are always **much better** than streamflow forecasts based on post-processed precipitation forecasts in system'. They don't give the readers an objective description for the research. The qualitative results cannot help the scientific choice. Please provide more numerical results, especially in conclusion.*

    **Authors Replay (AR):** We provided a more in depth numerical analysis in the revised version.

2. **RC1:** *Concerns about the multimodel approaches. From Figure 8, the multimodel approach seems to bring additional uncertainty to streamflow forecasts, as system C always has the worst results in term of BIAS and IQR. From Figure 5, the models are with an average KGEm of 0.64 in validation without EnKF. This is a quite low value. When the hydrological uncertainty is dominant, it is difficult to analyze the effect from precipitation post-processor. So, the boxplot for system C with or without post-processor for lead time 1 day in Figure 11 is similar. It is not indicated that the precipitation post-processor brings in no improvements. The improvements probably are too minor to offset the hydrological errors.*

    **AR:** This is a very interesting point, and we thank the reviewer for drawing attention to it.

    The 0.64 is the mean of all models in all catchments considered separately, not as a multimodel. That is, the performance of each of the models was determined individually and then averaged. In the case of the multimodel, its average yield is 0.73 (see figure attached to question 12 of reviewer #2). The simulations of each model are considered as an ensemble and then their performance is determined.
    It is also good to remember that extreme values affect the mean. Various models experienced difficulties in simulating some basins, which is to be expected since no single model excels in all situations. In the case of systems using only one model, the one used is the median during

calibration. As we responded to reviewer 2, it is almost impossible to predict which model will be the best predictor on any given day and basin, and that is when a multimodel has value.

We added the performance of the multimodel in the complementary material to avoid confusion, and we also softened our wording to recognize the value of the post-processor. This point raised by the reviewer is worth mentioning, and we included a comment on it.

3.  **RC1:** *The author's language usage was difficult to read at times. Too many adverbial clauses and attributive clauses make the sentence too long to understand.*

   **AR:** Fixing this was a priority when producing the revised version.

4. **RC1:** *In Section 2.1.1, the authors reduced the ECMWF database to 0.1° and then spatially averaged forecasts to the catchment scale. I am confused by the resolution reduction as the catchment areal forecasts were used. It is more simple to use the archived 0.25° to calculate the catchment average. The resolution reduction might bring in additional uncertainty to precipitation forecasts.*

   **AR:** We fully agree with the reviewer that resolution reduction might bring additional uncertainty to precipitation forecasts. However, the original resolution of the ECMWF is too coarse for this application, especially for the smaller group of catchments (11 in total). The surface of this group is less than 800 km2, and in many cases, only one meteorological forecast grid point falls within catchment boundaries and in others none. Therefore, downscaling ensures that several points fall within the catchment boundaries.

    In addition, Thiboult et al. 2016 suggested that when we reduce the spatial resolution, we consider the contribution of the points close to the catchment boundaries, which allows us to have a better description of the meteorological conditions of the catchments. This is also corroborated by Scheuerer and Hamill 2015a who demonstrated that it is beneficial to add forecasts from grid points within a certain neighborhood of the location of interest as potential predictors to account for position uncertainty.

   We clarified the purpose behind the downscaling in the revised version.

5. **RC1:** *Line 170-171: The authors mentioned they used an adapted CSGD to post-process ECMWF precipitation. Whether the only difference is that the original CSGD used neighboring grid points while they used all grids in a catchment?*

   **AR:** The reviewer understood this issue correctly. The original CSGD uses neighborhood information from grid-based forecasts to compute the ensemble statistics, thus accounting for displacement errors. In our case, we computed the ensemble statistics directly from the mean

areal precipitation over catchments. As mentioned in the previous answer, the downscaling implicitly allows considering the contribution from neighboring grids.

6. **RC1:** *Line 221-222: the same….as….*

    **AR:** Corrected

7. **RC1:** *Line 506-509: NWP products often fail to capture precipitation forecasts in small domain, yet behave better in large catchments. Lumped hydrological models are more likely to better model streamflow at small basins, where the hydrological process is simpler and easier to be simulated by those simple lumped models.*

    **AR:** Thank you for pointing this out. As suggested by Reviewer # 2, we shifted the emphasis to the properties of the catchments rather than the modeling approach (see our answer to question 13 of Reviewer #2 for details).

8. **RC1:** *Since the authors elaborately analyze the performance for different catchment size in Section 3.4, they should provide detailed validation results of the hydrological models in different catchments in Section 3.1. It would help the readers to understand the forecast skill over catchment sizes.*

    **AR:** Thank you very much for this suggestion. We included the figure below and moved Section 3.1 to the supplementary material as recommended by Reviewer #2 to decrease the length of the paper.

[Figure]

**Figure 3.** Performance of the seven hydrological models in terms of volumetric efficiency (VE), KGEm, and relative bias (BIAS) in calibration (1997-2007; red), validation (2008-2016) without EnKF DA (light blue), and with EnKF DA (dark blue). Boxplots represent the distribution of the scores over the catchments in each group: Smaller, Medium and Larger. The scores exclude wintertime (December-March).

**RC2: Anonymous Referee #2**

**Major comments:**

1. **_RC2:_** _Figure 8, and the commentary associated with it, is problematic. The authors are comparing reliability of their hydrological uncertainty quantification methods when they know that their forcings are unreliable (as deftly shown in fig 10). This makes assessments of reliability meaningless: the streamflow forecasts in fact *should* be unreliable, as the forcings are unreliable. So, for example, in Fig 8 System B looks bad next to System D (for reliability), but Fig 13 shows that when forcings are reliable, System B performs similarly to System D. So Fig 8 gives misleading information - indeed, one could argue it shows System B to be superior (despite the appearance to the contrary). There are a few ways around this - e.g. the authors could generate hydrological 'forecasts' that are forced with observations to produce Fig 8 - but the simplest way is to remove this figure and the accompanying discussion, and focus more on Figs 11-14._

   **Authors Replay (AR):** This is a relevant remark, and we understand the reviewer's point. Unfortunately, current operational systems are often like systems A, B, and C. We wanted to overview how each tool contributes to the forecast performance over time and how a post-processor can improve it. This highlights three important aspects: 1) the need to correct the inputs before being used, 2) how reliability increases as we quantify other sources of uncertainty, and 3) the fact that systems like Sys-A could still lead to non-satisfactory forecast quality performance. Furthermore, as suggested, we used the observations to explore a new figure similar to Fig. 8, and the conclusions are the same (Sys-A is the least reliable).

   We removed Fig. 8 but to shorten the length of the paper. The comments associated with it are replicated in Figs. 6 and 7 of the revised version.

2. **_RC2:_** _The paper is admirably comprehensive, but perhaps as a result somewhat long. Longer papers are less likely to be read, and I really feel like this paper deserves to be read widely! So I have a few suggestions for shortening it:_

   **AR:** Thank you for your comments that greatly contribute to improving our paper.

   _i) Figure 7 replicates information in Figure 9. I suggest omitting Figure 7 and restructuring the paper to first discuss rainfall forecasts (both raw and post-processed) and then streamflow forecasts._

   **AR:** Following the reviewer's suggestion, we changed the structure of the result section as follows (Section 3):

   3. Results
      3.1 Ensemble precipitation forecasts
         3.1.1 Performance of raw precipitation forecasts
         3.1.2 Performance of corrected precipitation forecasts

*ii) Fig 8 should be omitted, as noted above*

**AR:** See the answer to question 1.

*iii) Figs 11-14 could be combined into a single figure. At the very least, I suggest Figs 13 & 14 should be combined - as the authors note, sharpness without reliability is meaningless.*

**AR:** We thank the reviewer for the suggestion. We used the following figures in the revised version:

[Figure]

**Figure 6.** Relative bias (BIAS) and MCRPS of the ensemble streamflow forecasts of the four hydrological prediction systems and lead times 1, 3, and 6 days when considering raw (blue) and post-processed (red) precipitation forecasts. Boxplots represent the score variability over the 30 catchments.

[Figure]

**Figure 7.** MAE of the Reliability Diagram (MAErd) and Interquartile Range (IQR) of the ensemble streamflow forecasts of the four hydrological prediction systems and lead times 1, 3, and 6 days when considering raw (blue) and post-processed (red) precipitation forecasts. Boxplots represent the score variability over the 30 catchments.

*iv) [Optional] The discussion of the hydrological model performance (Section 3.1) and the accompanying figure, while interesting, could be moved to supplementary materials.*

**AR:** Thank you for this suggestion. We moved Section 3.1 to supplementary material and added another figure with detailed validation results of the hydrological models for the different catchment groups as suggested by reviewer #1.

3. **_RC2:_** *Figure 15 shows results that appear to be somewhat inconsistent with Figs 12 & 13. For example, in Figure 15 Sys-C looks best (most reliable) across all catchments compared to other systems - much better than Sys-B or Sys-D. This is very much not the case in Fig 13, where Sys-B & Sys-D are more reliable when paired with the CSGD forcing. Further, the RMSE values for the CSGD forcings are often almost identical to the raw forcings, which is not consistent with the CRPS scores presented in Fig 12. I think spread-skill plots are a pretty rough way of looking at reliability (they reduce the 'spread' of the ensemble to root of the average variance, rather than considering the full distribution, e.g. like Probability Integral Transforms/Rank Histograms do), and I suspect the reliability diagram analyses presented by the authors in Fig 13 give a better picture. It may be better to simply use MAE_rd & CRPS in Figure 15 for consistency with Figs 12 and 13. If the authors keep the spread-skill plots in Fig 15 (they may have different views to me on the efficacy of the different reliability assessments) - I think the apparent discrepancies between Fig 15 and Figs 12/13 should be explained.*

**AR:** This is a relevant remark. We followed the reviewer's suggestions and used the MAErd for Fig. 15 (Fig. 8 in the revised version). In this way, we avoided the multiplication of indicators that sometimes react differently. Therefore, we are now consistent with the other figures.

However, it is good to note that in Fig. 15, we evaluated reliability by catchment group (lines presented the median) as opposed to Figs. 11-14, where the boxplots included all watersheds.

On the other hand, the RMSE was larger in the raw than in the corrected forecasts in the medium catchment group. Moreover, the RMSE value was larger in system C than in the others. This was congruent with Fig. 12. There should indeed be some congruence between the MCRPS and RMSE, but both evaluate different attributes of the forecast quality.

**Specific (minor) comments:**

1. **RC2:** L26-27 'The inherent uncertainty of hydrological forecasts stems from three main sources' there is a crucial fourth source the authors have omitted: uncertainty in observations.

   **AR:** Definitively, the reviewer is right. We added this source of uncertainty in the revised version.

2. **RC2:** L28 "Two main philosophies are generally adopted" I understand why the authors are making this distinction, but I guess I would argue the emergence of a third class of systems, namely hybrid statistical-dynamical forecasting systems - which combine physical models (e.g. ensemble NWP) with statistical processing methods (e.g. NWP calibration) - is worth mentioning. Statistical post-processors (hydrological uncertainty processors, ensemble dressing) are very effective at correcting statistical aspects of forecasts (bias, uncertainty quantification), but usually produce discrete probability distributions in time (e.g. one for each lead time) and space (e.g. one for each gauge) - they are not connected in time and space. Users of hydrological forecasts are often interested in hydrographs, which require coherent information in time (across lead times) and (often) in space (e.g. upstream and downstream gauges). Coherent spatial and temporal information allows users to, for example (i) check flood peaks as well as total volume forecasts for, say, the next week; (ii) when flood peaks are expected to pass by a sequence of gauges. Ensemble systems can present all this information, but when they are based on purely physical/conceptual models they often do not get statistical aspects of forecasts (bias, uncertainty quantification) correct. Hybrid statistical-dynamical systems are in their infancy, but they promise the best of both purely statistical or purely dynamical ensmeble systems.

   **AR:** Thank you very much for the suggestion. We included this valuable information in the new version.

3. **RC2:** L71-73 "Hydrological post-processing ... cannot compensate for weather forecasts that are highly biased" Yes, it can - though it depends on the method employed. Hydrological uncertainty processors (e.g. Biondi and Todini 2018) lump all sources of uncertainty - including biases in rainfall forecasts - into the model, and they are effective at correcting these issues. They are not

*so good if you need each ensemble member to represent a temporally coherent hydrograph, however.*

**AR:** The reviewer is right. Some hydrological post-processing techniques have been developed to directly address the system's total uncertainty (Demirel et al.2013, Kavetski et al.2006), but this is a subject still in its infancy and, to our knowledge, have not been implemented in an operational context. In many cases, the results are synthetic series that represent a probability and not a real forecast. As the reviewer points out here and in question 2, the temporal representation of each member may be important if the ultimate goal is to feed another model (e.g., hydraulic). This lack of consistency is likely to generate a different response from the catchment (faster/slower flood, event duration, etc.) and is not desirable. We added this in the revised version:

"Hydrological post-processing … cannot compensate for weather forecasts that are highly biased (Abaza et al., 2017; Roulin and Vannitsem, 2015) unless it addresses the many sources of uncertainty in an integrated manner (Biondi and Todini 2018, Demirel et al.2013, Kavetski et al.2006)."

4. **RC2:** *L84 "Considering several sources of uncertainty in ensemble hydrometeorological forecasting inevitably implies an increase in the number of simulations" This is not 'inevitable' - e.g. Bennett et al. (2016), who add hydrological uncertainty to rainfall forecast uncertainty without increasing the number of ensemble members. Suggest this be changed to "Considering several sources of uncertainty in ensemble hydrometeorological forecasting often implies an increase in the number of simulations".*

**AR:** Thank you for the remark. We modified this in the revised version.

5. **RC2:** *L87 "However, there is also additional uncertainty associated with the assumptions made in creating a larger ensemble." Please be explicit about why increasing ensemble size is a problem.*

**AR:** In this case, we refer to ontological uncertainty (uncertainty associated with different belief systems). As highlighted by (Beven, 2016), different beliefs about the appropriate assumptions could lead to very different uncertainty estimates. Therefore, we modified the paragraph as follows:

"However, there is also additional uncertainty associated with the assumptions made in creating a larger ensemble. A plethora of methods exists to quantify each source of uncertainty, which implies ontological uncertainties (Beven, 2016). If the methodologies implemented to simulate the sources of forecast error and quantify various uncertainty sources are inappropriate, even a large ensemble size could be under or over dispersive and thus not represent the total predictive uncertainty accurately (Boelee et al., 2019; Buizza et al., 2005)."

6. **RC2:** *L137 "modeling uncertainty" should this be "hydrological modeling uncertainty"?*

**AR:** We corrected this in the revised version.

7. **RC2:** *L211-213 "As such, it relaxes the constraint of parametric assumptions (difficult for variables like precipitation) and high dimensional cases (Gneiting, 2014). This technique uses the raw ensemble forecasts trajectories to identify the dependence template. Therefore, the post-processed ensemble should have the same number of members as the raw ensemble." I didn't find this explanation very clear or helpful. ECC enforces rank correlations (both temporal and spatial) from the raw forecasts in the calibrated forecasts, much like the Schaake shuffle. I would simply say something like this.*

    **AR:** Thank you very much for this suggestion. Corrected to the following:

    "This technique reconstructs the spatio-temporal correlation *by reordering samples from the raw predictive marginal distributions* to identify the dependence template. The ECC reasoning lies in the fact that the physical model can adequately represent the covariability between the different dimensions (i.e., space, time, variables). Furthermore, as the template is the raw ensemble, the ECC should have the same number of members as the raw ensemble."

8. **RC2:** *L219 "provide model state" The authors use a range of models (nwp, statistical, hydrological), and it would be helpful to clearly identify (here and elsewhere) which ones they are talking about in each case, rather than referring to generic 'models'. Here I think the authors are discussing hydrological model states.*

    **AR:** The reviewer is right, in L219 we were discussing hydrological model states. We identified which model we were referring to throughout the paper in the revised version.

9. **RC2:** *L315 "Root Mean Square Error" is this calculated on the ensemble mean? (And if not, how is it calculated?)*

    **AR:** Yes, it was calculated on the ensemble mean. However, as suggested by the reviewer, we made Fig. 15 (now Fig. 8) based on the MAErd to shorten the paper and maintain consistency with the other figures. All information related to the spread-skill plot was removed in the revised version.

10. **RC2:** *L333-334 "Accordingly, we decided to use the reliability diagram to evaluate precipitation for thresholds of different exceedance probabilities (EP), namely 0.05, 0.5, 0.75, and 0.95". Could the authors please confirm 1) that to construct the reliability diagrams, they have converted probabilistic forecasts to binary predictions for exceeding thresholds based on these quantiles and 2) these quantiles are taken from observations and 3) that they have calculated MAE_rd across all these thresholds? These things weren't entirely clear to me from the description. L338 IQR (interquartile range) usually denotes the range between two quartiles (25%ile and 75%ile). So 'IQR' is not the correct term to use for a different interval. Suggest using the term 'average width of prediction interval' (AWPI).*

1) Yes. The reviewer is correct.

2) Yes, those quantiles were taken from the observations.

3) No, we just used those quantiles for Figure 10 (Fig. 5 in the revised version). To compute the MAErd, we calculated nine confidence intervals with nominal confidence level of 10–90%, with an increment of 10% for each emitted forecast. Then, it was established whether or not each confidence interval covered the observation for each forecast and each confidence interval. This was repeated for all the forecast–observation pairs. In a well-calibrated distribution, the observation inside each confidence interval and its corresponding nominal confidence level should be close, taking the form of a linear relationship 1:1 (as the reliability diagram). Finally, the difference between these values is equivalent to the MAE of the reliability diagram.

In L338 it was a typo error. The correct word is Interquantile, as shown in Figure 3. We used the 90% Interquantile range, the difference between the 95th quantile and the 5th quantile (Gneiting et al. 2007) that have been used widely as a metric to evaluate the sharpness in the hydrometeorological forecasting community (Cassagnole et al. 2021, Bourgin et al. 2014, Crochemore et al., 2016).

We clarified all these points in the revised version.

11. **RC2:** *L362 Figure 4 - this is a really nice figure!*

    **AR:** Thank you! 😊

12. **RC2:** *L386 Figure 5 - would be helpful to have abbreviated names of the models along the xaxis rather than M1, M2, etc, to avoid having to track back to Table 2 to interpret this figure. These only need to appear on the bottom panel.*

    **AR:** We used this figure in the revised version (caption remains the same):

[Figure]

13. **RC2:** *L507 "This is particularly observed in the streamflow forecasts and may be related to the use of lumped hydrological models, which tend to smooth errors over larger modeling areas" This is kind of saying the same thing, but I'd shift the emphasis to the properties of the catchments themselves, rather than lumped hydrological models. Something like "For a given time step, smaller catchments tend to show more variance and are thus more difficult to forecast than larger catchments." (I.e., it's the larger catchments themselves that smooth, e.g., spatial and temporal variability in rainfall; the models just try to replicat this.)*

**AR:** We thank the reviewer for the suggestion. Reviewer #1 also pointed out this. We modified it accordingly in the revised version. Corrected to the following:

"This is particularly observed in the streamflow forecasts and may be related to the fact that larger catchments tend to experience lower streamflow variability (Sivapalan, 2003), and it is thus easier for the hydrological models to simulate their streamflows (Andréassian et al., 2004). Moreover, weather can differ substantially over a couple of kilometers, and the resolution of NWP models is often too coarse to capture these variations in smaller catchments. For example, extreme localized events can be missed in small catchments if the amount of precipitation is well predicted but in the wrong location. In larger catchments, a buffer effect can be generated, and displacements of precipitation may impact less the predictions of streamflow."

14. **RC2:** *L515-516 "These findings confirm that large improvements in precipitation forecasts do not necessarily lead to improvements in streamflow forecasts." I don't wholly agree with this. First,*

*Sys-B \*does\* show marked improvement in reliability when it is forced with reliable inputs. So there is improvement there. The 2 best peforming methods, to my eye, are Sys-B and Sys-D when forced with Q_CSGD. Sys- B is marginally superior: at Day 6, Sys-B with Q_CSGD is sharper than Sys-D, despite being similarly reliable. (When you consider the additional complexity of Sys-D, Sys-B to my mind is clearly preferable for operational forecasting.) Further, the measures of reliability the authors have chosen may not be all that sensitive. I personally prefer the use of probability integral transforms (PIT), or their summary statistics (e.g. Renard et al.'s (2010) 'alpha' statistic - very similar to the authors' MAE_rd). Because PIT treats forecasts as continuous variables, there's no need to simplify forecasts to binary predictions of thresholds. I'm not suggesting any changes here - the analysis the authors have presented is easily thorough enough, but I think they are understating the value of the CSGD in the system, and should probably soften their wording here.*

*__AR:__ In L515-516 we referred particularly to smaller catchments, not in a general sense. Improvements brought by the post-processor were more significant in terms of RMSE in precipitation than in terms of Spread. The opposite was true for streamflow. We recognize the value of CSGD. That is why we generally said: "not necessarily." In fact, in Table 4 (now Table 5), we highlight the systems that, together with the post-processor, could be equal or better than system D. We conclude that the effectiveness is conditioned to several factors, and one of them is the catchment's size. We emphasize the value of the CSGD in the corrected version (see the new Table 4, for instance).*

15. __RC2:__ *L537 "In system D, the application of precipitation post-processing does not lead to reliable streamflow forecasts: the RMSE and spread curves in red in Fig. 15 are not aligned, while the blue curves (for raw forecasts) are for lead times greater than 3 days" The reason seems clear: Sys-D ensembles are wider than they need to be at Day 6, as shown cf Sys-B in Fig 14. So while they are reliable, they are not as sharp as they could be. In short, Sys-D overestimates uncertainty at long lead times.*

    *__AR:__ The reviewer is correct. But as mentioned above, all spread-skill plot related figures and analyses were removed in the revised version.*

16. __RC2:__ *L574 Fig 16 - this is a fascinating figure, probably worth a paper on its own. Great stuff!*

    __AR:__ Thank you! 😊

17. __RC2:__ *L578 "(Fig 15 system C)" As already noted, I have misgivings about Fig 15. In other figures (Figs 11-14) Sys-C is clearly outperformed by both Sys-B and Sys-D. And the reason is not really*

surprising: hydrological models, while somewhat diverse in structure (in some respects they can be quite similar, too), often perform very similarly once calibrated (as the authors themselves show in Section 3.1). So the uncertainty explained by a range of calibrated models is likely to be quite small, explaining why Sys-B can outperform Sys-C even though it uses a single hydrological model.

**AR:** *We do not wholly agree with this. Figure 5 showed that the models react differently from one location to another (boxplot variability). Models 2, 3, and 5 show a similar range, but this is not the case for models 1, 4, 6, and 7. Numerous papers confirm that parameter uncertainty is accounted for when a multimodel approach is used. The figure attached to question 12 makes this clear. We can see that a multimodel approach is more beneficial than using models 1, 4, 6, and 7 individually. Some models simulate some events better than others, so their performance varies during a hydrological year. We know that it is a nightmare and almost impossible to predict which model will be the best predictor on any given day, and that is when the multimodel approach has its value.*

18. **RC2:** *L581 "If the priority is to achieve a reliable and accurate system, then system B with precipitation post-processor presents a better alternative than a system like D." Yes, I fully agree.*

   **AR:** Thank you! 😊

19. **RC2:** *L586 "When post-processing is not applied to precipitation forecasts... systems with multimodel provide better reliability" I agree that this is what Figs 8 and 15 show, but as discussed: Fig 8 shouldn't be expected to produce reliable forecasts, and Fig 15 appears to be at least partly inconsistent with Figs 12 & 13. These conclusions can only be drawn with certainty if Fig 8 was constructed from streamflow forecasts generated with 'perfect' (observed) rainfall forcings to isolate the effects of the hydrological uncertainty generation methods. I don't think this paper needs this additional analyses - it has plenty of material already - so I suggest omitting this paragraph.*

   **AR:** *As mentioned in question 1, even using perfect forecasts, the conclusions remain the same. The hydrological ensemble variation is substantially larger and contributes more actively over time to reliability than the meteorology or initial conditions ensembles.*

20. **RC2:** *L589-590 "System B, with 2,500 members, performed similarly to system C (350 members) in terms of MCRPS as the EnKF effect fades with increasing lead times." I hadn't realised the difference in ensemble size was so stark. It's likely that the discrepancy in ensemble sizes would have impacted on all their calculations (though I accept the authors' contention that the effect*

*was small) - however for future work I suggest subsampling from the large ensembles to control for ensemble size, or using unbiased estimators that control for ensemble size (e.g. Ferro et al. (2008)) so any potential artifacts of ensemble size are totally removed.*

**AR:** We are aware that the ensemble size may generate bias implying that the apparent superiority of a system is due solely to its larger ensemble size and not to differences in modeling. Most studies conclude that increasing the ensemble size improves performance. However, that was not always the case in our study. Moreover, as pointed out by Buizza and Palmer 1998, performance improvement brought by increasing the ensemble size is highly dependent on the metric used to evaluate performance.

We explored the correction factor proposed by Ferro et al. 2008 but seeing that the difference was not substantial (the difference was in the 3rd decimal), we decided to use the standard MCRPS to emphasize that the technique used is more significant than the ensemble size, as expressed by Sharma et al. 2019.

21. **RC2:** *L630 "Since model C presented a good option, it would be interesting to determine if this system with a hydrological post-processor that corrects the models' bias would improve the forecasting performance without resorting to sophisticated precipitation post-processor techniques." To reiterate, the conclusion that System C is a good option is really based only on Fig 15, about which I have misgivings. If these misgivings are borne out, I would suggest that System C would not necessarily be considered a 'good option'.*

**AR:** *It is true that in most of the forecast attributes evaluated, system C presented a lower performance than systems B and D. However, system C presented a better MAErd than system B, especially on the farthest horizons. Perhaps "good option" is not appropriate to use, and it should be specified that this is in terms of reliability. We modified it in the revised version as follows:*

*"Since systems with multimodel provide better reliability, it would be interesting to determine if this system with a hydrological post-processor that corrects the models' bias would improve the forecasting performance without resorting to sophisticated precipitation post-processor techniques."*

**Typos etc.**

*L38 "parameters uncertainty" should be "parameter uncertainty"*
Corrected in the revised version

*L101 "directly use NWPs outputs" should be "directly use NWP outputs"*

Corrected  as well  in the revised version

**References**

Sivapalan, M. (2003), Process complexity at hillslope scale, process simplicity at the watershed scale: Is there a connection?, Hydrol. Processes, 17, 1037–1041, doi:10.1002/hyp.5109.

Kavetski D, Kuczera G, Franks SW. Bayesian analysis of input uncertainty in hydrological modeling: 2. Application. Water Resour Res 2006, 42:W03408. https://doi.org/10.1029/2005WR004376

Demirel MC, Booij MJ, Hoekstra AY. Effect of dif- ferent uncertainty sources on the skill of 10 day ensemble low flow forecasts for two hydrological models. Water Resour Res 2013, 49:4035–4053.

Gneiting, T., Balabdaoui, F., and Raftery, A.: Probabilistic forecasts, calibration and sharpness, J. Roy. Stat. Soc. B, 69, 243–268, https://doi.org/10.1111/j.1467-9868.2007.00587.x, 2007.

Crochemore, L., Ramos, M.-H., and Pappenberger, F.: Bias correcting precipitation forecasts to improve the skill of seasonal streamflow forecasts, Hydrology and Earth System Sciences, 20, 3601–3618, https://doi.org/10.5194/hess-20-3601-2016, 2016.

Zalachori, I., Ramos, M.-H., Garçon, R., Mathevet, T., and Gailhard, J.: Statistical processing of forecasts for hydrological ensemble prediction: a comparative study of different bias correction strategies, Advances in Science and Research, 8, 135–141, https://doi.org/10.5194/asr-8-135-2012, 2012.

Bourgin, F., Ramos, M. H., Thirel, G., and Andréassian, V.: Investigating the interactions between data assimilation and post-processing in hydrological ensemble forecasting, Journal of Hydrology, 519, 2775–2784, https://doi.org/10.1016/j.jhydrol.2014.07.054, 2014.

Cassagnole, M., Ramos, M.-H., Zalachori, I., Thirel, G., Garçon, R., Gailhard, J., and Ouillon, T.: Impact of the quality of hydrological forecasts on the management and revenue of hydroelectric reservoirs – a conceptual approach, Hydrology and Earth System Sciences, 25, 1033–1052, https://doi.org/10.5194/hess-25-1033-2021, 2021.